# Improving the Continuity of Goal-Achievement Ability via Policy Self-Regularization for Goal-Conditioned Reinforcement Learning

Xudong Gong [1 2]   Sen Yang [1]   Dawei Feng [1 2]   Kele Xu [1 2]   Bo Ding [1 2]   Huaimin Wang [1 2]   Yong Dou [1]

## Abstract

This paper addresses the challenge of discontinuity in goal-achievement capabilities observed in Goal-conditioned Reinforcement Learning (GCRL) algorithms. Through a theoretical analysis, we identify that the reuse of successful trajectories or policies during training can aid in achieving adjacent goals of achievable goals. However, the policy discrepancy between achievable and adjacent goals must be carefully managed to avoid both overly trivial and excessively large differences, which can respectively hinder policy performance. To tackle this issue, we propose a margin-based policy self-regularization approach that optimizes the policy discrepancies between adjacent desired goals to a minimal acceptable threshold. This method can be integrated into popular GCRL algorithms, such as GC-SAC, HER, and GC-PPO. Systematic evaluations across two robotic arm control tasks and a complex fixed-wing aircraft control task demonstrate that our approach significantly improves the continuity of goal-achievement abilities of GCRL algorithms, thereby enhancing their overall performance. Our code is available at https://github.com/GongXudong/fly-craft-examples.

## 1. Introduction

Goal-conditioned Reinforcement Learning (GCRL) (Liu et al., 2022) is an approach that acquires goal-conditioned behaviors by maximizing cumulative rewards over a desired goal distribution through trial-and-error (Sutton & Barto,

[1]College of Computer Science and Technology, National University of Defense Technology, Changsha, Hunan, China [2]State Key Laboratory of Complex & Critical Software Environment, Changsha, Hunan, China. Correspondence to: Dawei Feng <davyfeng.c@qq.com>.

*Proceedings of the $42^{nd}$ International Conference on Machine Learning*, Vancouver, Canada. PMLR 267, 2025. Copyright 2025 by the author(s).

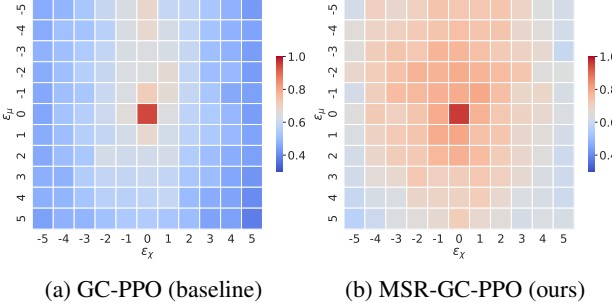

(a) GC-PPO (baseline)          (b) MSR-GC-PPO (ours)

*Figure 1.* Success rate of adjacent goals around an achievable goal on the fixed-wing aircraft control task. In this task, $\chi$ and $\mu$ represent two dimensions of the goal space. The central point in the figure denotes an achievable goal. To enhance the clarity of the two coordinate axes, we introduce the baseline noise parameters $\Delta_\chi$ and $\Delta_\mu$. The horizontal axis, $\epsilon_\chi$, signifies that the distance between an adjacent goal and the achievable goal along the $\chi$ dimension is $\epsilon_\chi \Delta_\chi$. A similar interpretation applies to the vertical axis, $\epsilon_\mu$. We train 5 policies, each initialized with a distinct random seed. For each policy, we conduct evaluations with 100 achievable goals. The results presented herein reflect the average performance across the total of 500 achievable goals.

2018). This method demonstrates superiority over conventional approaches of training a policy for each individual goal, as it facilitates the transfer of knowledge across different goals (Andrychowicz et al., 2017). GCRL has been extensively applied in control domains, such as robotic arm manipulation (Pitis et al., 2020) and fixed-wing aircraft control (Gong et al., 2024b; 2025b), as well as in games like MineCraft (Yuan et al., 2024) and Atari (Warde-Farley et al., 2018; Schaul et al., 2015).

However, in practice, we observe that GCRL algorithms commonly encounter discontinuity in their capability to achieve goals. For instance, when the error threshold for determining achievement, $\delta = 1$, an aircraft is capable of executing a right turn of 30 degrees, yet it fails to accomplish a right turn of 30.1 degrees. This is an issue that should not arise, given that the change in the goal, 0.1, is significantly less than $\delta$. Consequently, reusing the trajectory for the 30-degree right turn should suffice to achieve the 30.1-degree turn, as demonstrated in Fig. 2 (a) and (b). To provide a more intuitive illustration, we conduct systematic evalua-

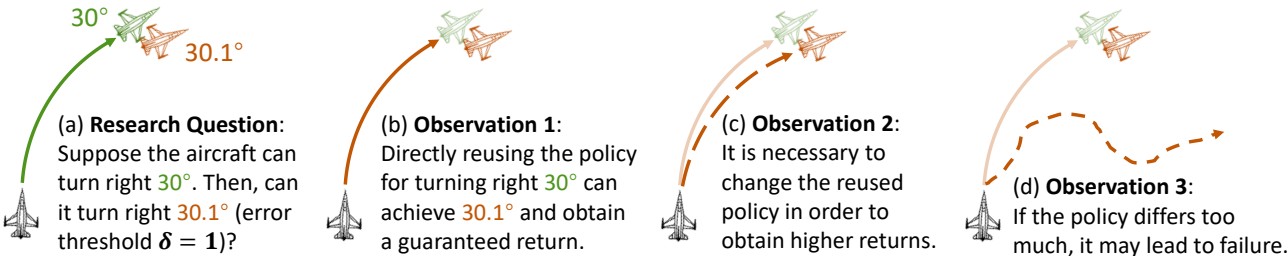

(a) **Research Question**: Suppose the aircraft can turn right 30°. Then, can it turn right 30.1° (error threshold $\delta = 1$)?

(b) **Observation 1**: Directly reusing the policy for turning right 30° can achieve 30.1° and obtain a guaranteed return.

(c) **Observation 2**: It is necessary to change the reused policy in order to obtain higher returns.

(d) **Observation 3**: If the policy differs too much, it may lead to failure.

*Figure 2.* Research question and three corresponding observational findings.

tions on the fixed-wing aircraft control task, and the results are depicted in Fig. 1. It is evident that the success rate for goals in the vicinity of an achievable goal typically falls below 60%. In contrast, our method significantly elevates the success rate to well above 60%.

To address the aforementioned issue, we conduct a theoretical analysis which reveals that, during training, reusing successful trajectories can facilitate the policy in learning to achieve adjacent goals around the achieved ones, while also guaranteeing a baseline level of cumulative rewards. The difference between policies for achieved goals and their adjacent goals should be small but not too trivial; an excessively small discrepancy may hinder the policy from further enhancing cumulative rewards beyond the baseline (Fig. 2 (c)), whereas a large discrepancy may result in the policy's failure to obtain the baseline cumulative rewards (Fig. 2 (d)).

Building upon these insights, we develop a **M**argin-Based Policy **S**elf-**R**egularization (MSR) approach, which optimizes the difference between the polices for a desired goal and its adjacent goals to a threshold that represents the minimal acceptable discrepancy. We incorporate MSR into the widely used GCRL algorithms, including the goal-conditioned off-policy algorithms GC-SAC (Pitis et al., 2020) and HER (Andrychowicz et al., 2017), as well as the goal-conditioned on-policy algorithm, GC-PPO (Gong et al., 2024a). We conduct experiments on two robotic arm control tasks (Gallouédec et al., 2021) and one complex, real-world fixed-wing aircraft control task (Gong et al., 2025a). Experiment results indicate that MSR effectively improves the continuity of goal-achievement abilities for GCRL algorithms, and furthermore, it improves the ability of the policies to obtain cumulative rewards.

To summarize:

- We assess multiple prevalent GCRL algorithms across a range of tasks, substantiating the prevalent challenge of discontinuity in the goal-achievement capabilities of GCRL algorithms.

- We conduct a theoretical analysis to elucidate the reasons for the discontinuity in goal-achievement capabilities of GCRL algorithms. Based on the insights from theoretical analysis, we design a margin-based policy self-regularization method. This method, serving as an auxiliary optimization for policies, can be integrated as a plug-in module into nearly all GCRL algorithms.

- We conduct systematic evaluations on two robotic arm control tasks and a fixed-wing aircraft control task. Results indicate that the integration of MSR into GCRL algorithms substantially improves the continuity of their goal-achievement abilities.

## 2. Preliminaries

**GCRL** can be described by goal-augmented MDP (Liu et al., 2022) $\mathcal{M} = \langle \mathcal{S}, \mathcal{A}, \mathcal{T}, r, \gamma, \mathcal{G}, p_{dg}, \phi \rangle$, where $\mathcal{S}, \mathcal{A}, \gamma, \mathcal{G}$ and $p_{dg}$ denote the state space, action space, discount factor, goal space and desired goal distribution of the environment, respectively. $\mathcal{T} : \mathcal{S} \times \mathcal{A} \to \mathcal{P}(\mathcal{S})$ is the transition function, where $\mathcal{P}(\mathcal{X})$ denote the probability distribution over a set $\mathcal{X}$. $r : \mathcal{S} \times \mathcal{A} \times \mathcal{G} \to \mathbb{R}$ is the goal-conditioned reward function. $\phi : \mathcal{S} \to \mathcal{G}$ is a tractable mapping function that maps the state to a specific goal. The objective of GCRL is to achieve goals via a goal-conditioned policy $\pi : \mathcal{S} \times \mathcal{G} \to \mathcal{P}(\mathcal{A})$ that maximizes the expectation of the cumulative rewards over the desired goal distribution $J(\pi) = \mathbb{E}_{a_t \sim \pi(\cdot|s_t,g), g \sim p_{dg}, s_{t+1} \sim \mathcal{T}(\cdot|s_t,a_t)} \left[ \sum_t \gamma^t r(s_t, a_t, g) \right]$.

Without loss of generality, we assume that: (1) In the transition function, a parameter $\delta$ is incorporated to assess the achievement of a goal. The criterion for determining that a state $s$ has achieved the goal $g$ is given by the condition $\|\phi(s) - g\| < \delta$, where $\| \cdot \|$ represents a distance metric. If this condition is satisfied, it is said that the state $s$ has achieved the goal $g$. (2) The reward function is bounded by $R_{\max}$, i.e., $|r(s,a)| \leq R_{\max}, \forall(s,a) \in \mathcal{S} \times \mathcal{A}$.

To facilitate later analysis, we introduce the discounted stationary state distribution $d_\pi(s) = (1-\gamma) \sum_{t=0}^{\infty} \gamma^t \Pr(s_t = s; \pi)$, and the discounted stationary state-action distribution

$\rho_\pi(s, a) = (1 - \gamma) \sum_{t=0}^\infty \gamma^t \Pr(s_t = s, a_t = a; \pi)$. Intuitively, discounted stationary state (state-action) distribution measures the overall *frequency* of visiting a state (state-action) (Xu et al., 2020). Their relationship $\rho_\pi(s, a) = \pi(a|s)d_\pi(s)$ holds for any policy $\pi$. Additionally, we introduce an alternative method for computing the cumulative rewards, given by $J(\pi) = \frac{1}{1-\gamma} \mathbb{E}_{(s,a)\sim\rho_\pi} r(s, a)$, where $\frac{1}{1-\gamma}$ is analogous to the effective planning horizon (Puterman, 2014).

To maintain the clarity of the formulas, we employ the notation $\pi_g$ as an abbreviation for $\pi(\cdot|\cdot, g)$. We denote the cumulative rewards that policy $\pi$ can obtain with respect to a desired goal $g$ as $J(\pi_g)$, which is defined as $J(\pi_g) = \mathbb{E}_{a_t\sim\pi(\cdot|s_t,g),s_{t+1}\sim\mathcal{T}(\cdot|s_t,a_t)} \left[\sum_t \gamma^t r(s_t, a_t, g)\right] = \frac{1}{1-\gamma}\mathbb{E}_{(s,a)\sim\rho_{\pi_g}} r(s, a, g)$.

## 3. Margin-Based Policy Self-Regularization

### 3.1. Theoretical Analysis

We commence by presenting Theorem 3.1 and Corollary 3.2, which underpin the insight depicted in Fig. 2(b).

**Theorem 3.1.** *Consider a goal-augmented MDP $\mathcal{M} = \langle \mathcal{S}, \mathcal{A}, \mathcal{T}, r, \gamma, \mathcal{G}, p_{dg}, \phi \rangle$, where the error threshold for determining goal achievement is $\delta$. Suppose we have a trajectory $\tau = (s_0, a_0, s_1, a_1, \cdots, s_T, a_T)$ satisfying $\phi(s_T) = g$. For an arbitrary perturbation $\epsilon \in \mathcal{G}$, if $\|\epsilon\| < \delta$, then $\tau$ may also be regarded as achieving $g + \epsilon$.*

The proof of Theorem 3.1 can be found in Appendix A.2. Theorem 3.1 shows that if a trajectory successfully achieves goal $g$, then it is also able to serve as a trajectory for achieving goals in the vicinity of $g$. Theorem 3.1 analyzes the properties of achieved goals from the perspective of trajectories. The following Corollary 3.2, on the other hand, analyzes the properties of the achieved goal from the perspective of the policy.

**Corollary 3.2.** *Consider a goal-augmented MDP $\mathcal{M} = \langle \mathcal{S}, \mathcal{A}, \mathcal{T}, r, \gamma, \mathcal{G}, p_{dg}, \phi \rangle$, where the error threshold for determining goal achievement is $\delta$. For a desired goal $g'$, suppose we have a policy $\pi$ that can achieve this goal (with the actually achieved goal being denoted as $g$). For any perturbation $\epsilon \in \mathcal{G}$ such that $\|\epsilon\| < \delta$, the policy is guaranteed to achieve the goal $g + \epsilon$ on the condition that it satisfies $\mathbb{E}_{s\sim d_{\pi_{g'}}} D_{KL}(\pi(\cdot|s, g'), \pi(\cdot|s, g + \epsilon)) = 0$.*

The proof of Corollary 3.2 can be found in Appendix A.3. Corollary 3.2 indicates that if the policy remains unchanged, the same policy can also achieve goals in the vicinity of achieved goals.

Theorem 3.1 and Corollary 3.2 demonstrate that the trajectories or policies associated with achieved goals can be reused to achieve goals in the vicinity of these achieved goals. Theorem 3.3, meanwhile, attempts to quantify the cumulative rewards that can be obtained by reusing successful trajectories and policies.

**Theorem 3.3.** *Consider a goal-augmented MDP $\mathcal{M} = \langle \mathcal{S}, \mathcal{A}, \mathcal{T}, r, \gamma, \mathcal{G}, p_{dg}, \phi \rangle$. For any goal $g \in \mathcal{G}$ and perturbation $\epsilon \in \mathcal{G}$, the following two inequalities hold:*

$$
\begin{aligned}
J(\pi_{g+\epsilon}) \geq & \frac{1}{1-\gamma} \sum_{(s,a)} \rho_{\pi_g}(s, a) r(s, a, g + \epsilon) \\
& - \frac{2\sqrt{2}R_{max}}{(1-\gamma)^2} \sqrt{D_{KL}(\pi_{g+\epsilon}(\cdot|s), \pi_g(\cdot|s))}
\end{aligned} \tag{1}
$$

$$
\begin{aligned}
J(\pi_{g+\epsilon}) \leq & \frac{1}{1-\gamma} \sum_{(s,a)} \rho_{\pi_g}(s, a) r(s, a, g + \epsilon) \\
& + \frac{2\sqrt{2}R_{max}}{(1-\gamma)^2} \sqrt{D_{KL}(\pi_{g+\epsilon}(\cdot|s), \pi_g(\cdot|s))}
\end{aligned} \tag{2}
$$

The proof of Theorem 3.3 can be found in Appendix A.4. Theorem 3.3 establishes the upper and lower bounds for $J(\pi_{g+\epsilon})$. Here, $\frac{1}{1-\gamma} \sum_{(s,a)} \rho_{\pi_g}(s, a) r(s, a, g + \epsilon)$ represents the first part of return determined by the state-action distribution induced by $\pi_g$ and the reward function defined by $g + \epsilon$, while $\frac{2\sqrt{2}R_{max}}{(1-\gamma)^2} \sqrt{D_{KL}(\pi_{g+\epsilon}(\cdot|s), \pi_g(\cdot|s))}$ denotes the second part of return that is determined by the discrepancy between $\pi_g$ and $\pi_{g+\epsilon}$.

As indicated by Equation 1, when $\pi_g$ and $\pi_{g+\epsilon}$ are identical, $J(\pi_{g+\epsilon})$ obtains the maximum lower bound, $\frac{1}{1-\gamma} \sum_{(s,a)} \rho_{\pi_g}(s, a) r(s, a, g + \epsilon)$. We refer to this value as the **guaranteed return** for $\pi_{g+\epsilon}$. The guaranteed return corresponds to the cumulative rewards obtainable from reusing trajectories or policies as discussed in Theorem 3.1 and Corollary 3.2.

According to Equation 2, to enhance the return of $\pi_{g+\epsilon}$ beyond the guaranteed return, there must be a certain degree of discrepancy between $\pi_{g+\epsilon}$ and $\pi_g$ (as depicted by Fig. 2(c)). In the following section, we present Theorem 3.4 to elucidate the relationship between the policy discrepancy between $\pi_{g+\epsilon}$ and $\pi_g$ and the corresponding return gap.

**Theorem 3.4.** *Consider a goal-augmented MDP $\mathcal{M} = \langle \mathcal{S}, \mathcal{A}, \mathcal{T}, r, \gamma, \mathcal{G}, p_{dg}, \phi \rangle$. For a policy $\pi$ and any pair of goals $g_1 \in \mathcal{G}$ and $g_2 \in \mathcal{G}$, the following inequality holds:*

$$
\begin{aligned}
|J(\pi_{g_1}) - J(\pi_{g_2})| \leq & \\
\frac{1}{1-\gamma} \sum_{(s,a)} & \rho_{\pi_{g_1}}(s, a) |r(s, a, g_1) - r(s, a, g_2)| \\
& + \frac{2\sqrt{2}R_{max}}{(1-\gamma)^2} \mathbb{E}_{s\sim d_{\pi_{g_1}}} \sqrt{d_{KL}(\pi_{g_1}(\cdot|s), \pi_{g_2}(\cdot|s))}
\end{aligned} \tag{3}
$$

The proof of Theorem 3.4 can be found in Appendix A.5. Theorem 3.4 demonstrates that the gap in cumulative rewards between two goals, $g_1$ and $g_2$, under policy $\pi$ is bounded by two components: the first part is the return gap determined by the state-action distribution induced by $\pi_{g_1}$ and the reward gap determined by $g_1$ and $g_2$; the second part is the return gap determined by the discrepancy between $\pi_{g_1}$ and $\pi_{g_2}$. In the context of multi-goal problems, a commonly employed reward function is $r = -\|\phi(s) - g\|$. For this type of reward, we further simplify Equation 3 through Corollary 3.5.

**Corollary 3.5.** *Consider a goal-augmented MDP $\mathcal{M} = \langle \mathcal{S}, \mathcal{A}, \mathcal{T}, r, \gamma, \mathcal{G}, p_{dg}, \phi \rangle$, where the reward function is defined as $r(s, a, g) = -\|\phi(s) - g\|$. For a policy $\pi$ and any pair of goals $g_1 \in \mathcal{G}$ and $g_2 \in \mathcal{G}$, the following inequality holds:*

$$
\begin{aligned}
&|J(\pi_{g_1}) - J(\pi_{g_2})| \leq \\
&\frac{\|g_1 - g_2\|}{1 - \gamma} + \frac{2\sqrt{2}R_{max}}{(1-\gamma)^2} \mathbb{E}_{s \sim d_{\pi_{g_1}}} \sqrt{d_{KL}(\pi_{g_1}(\cdot|s), \pi_{g_2}(\cdot|s))}
\end{aligned}
\tag{4}
$$

The proof of Corollary 3.5 can be found in Appendix A.6. It can be observed that the first part of the return gap is reduced to $\frac{\|g_1 - g_2\|}{1-\gamma}$, a value that is solely determined by the difference between $g_1$ and $g_2$. When applied to our research problem, considering goals $g$ and $g + \epsilon$, we further simplify Equation 4 through Corollary 3.6.

**Corollary 3.6.** *Consider a goal-augmented MDP $\mathcal{M} = \langle \mathcal{S}, \mathcal{A}, \mathcal{T}, r, \gamma, \mathcal{G}, p_{dg}, \phi \rangle$, where the reward function is defined as $r(s, a, g) = -\|\phi(s) - g\|$ and the error threshold for determining goal achievement is $\delta$. For a policy $\pi$, a specific goal $g$, and any perturbation $\epsilon \in \mathcal{G}$, if $\|\epsilon\| < \delta$, the following inequality holds:*

$$
\begin{aligned}
&|J(\pi_g) - J(\pi_{g+\epsilon})| \leq \\
&\frac{\delta}{1 - \gamma} + \frac{2\sqrt{2}R_{max}}{(1-\gamma)^2} \mathbb{E}_{s \sim d_{\pi_g}} \sqrt{d_{KL}(\pi_g(\cdot|s), \pi_{g+\epsilon}(\cdot|s))}
\end{aligned}
\tag{5}
$$

The proof of Corollary 3.6 can be found in Appendix A.7. It is evident that for a goal $g$ and a goal within its $\delta$-ball, $g + \epsilon$, the return gap under policy $\pi$ is governed by two parts: a constant part determined by $\delta$ and a variable part determined by the discrepancy between $\pi_g$ and $\pi_{g+\epsilon}$. Current GCRL algorithms generally lack constraints on the discrepancy between policies for adjacent goals, which leads to the observed discontinuity in the policy's capability to achieve goals (as depicted by Fig. 2(d)). Consequently, Corollary 3.6 indicates that to reduce the return gap of a policy on two adjacent goals, it is necessary to diminish the policy discrepancy between these two goals.

## 3.2. The Proposed Method

We commence by summarizing the conclusions presented in Section 3.1:

- Corollary 3.2 indicates that a policy corresponding to an achieved goal can be reused to achieve goals in the vicinity of this achieved goal.

- Theorem 3.3 suggests that reusing a policy ensures a guaranteed return, yet higher returns can only be attained through policy modification.

- Corollary 3.6 demonstrates that when modifying the reused policy, the extent of modification should not be excessive, as this may result in the policy's return being unbounded.

Inspired by the aforementioned three conclusions, we are led to consider the reduction of the discrepancy between $\pi_g$ and $\pi_{g+\epsilon}$ while simultaneously preventing their convergence to excessive similarity. Consequently, we propose the MSR method, as delineated in Equation 6. The core component of this regularization involves the divergence between $\pi$ at $g$ and $g + \epsilon$, quantified by $D_{KL}[\pi_g(\cdot|s), \pi_{g+\epsilon}(\cdot|s)]$. Hence, the primary function of MSR is to mitigate the divergence between $\pi_g$ and $\pi_{g+\epsilon}$. The role of the parameter $\beta$ and the maximization operator is to ensure that the MSR ceases to exert influence when $D_{KL}[\pi_g(\cdot|s), \pi_{g+\epsilon}(\cdot|s)]$ is less than $\beta$, thereby preventing excessively small discrepancies between $\pi_g$ and $\pi_{g+\epsilon}$.

$$
L_{MSR}(\pi) = \max\{\mathbb{E}_{\substack{s \sim d_{\pi_g} \\ \epsilon \sim (-\epsilon', \epsilon')}} D_{KL}[\pi_g(\cdot|s), \pi_{g+\epsilon}(\cdot|s)] - \beta, 0\}
\tag{6}
$$

Given that MSR serves as a regularization for policies, it can be integrated with any policy-based GCRL method, whether it is off-policy, such as GC-SAC, or on-policy, such as GC-PPO. The specifics of this integration are detailed in Equation 7.

$$
L(\pi) = L_{RL}(\pi) + \lambda L_{MSR}(\pi),
\tag{7}
$$

where $L_{RL}(\pi)$ represents the optimization objective on $\pi$ in reinforcement learning and $\lambda$ is the strength of MSR. Algorithm 1 delineates the specific procedure for computing the MSR loss. It is important to note that for each transition, we sample $N$ perturbations, and based on Equation 6, we calculate $N$ MSR values. The average of these $N$ MSR values is then taken as the MSR for that particular transition. Increasing the value of $N$ can make the estimation of Equation 6 more accurate, however, it also leads to an increase in computational expense.

**Algorithm 1** Margin-Based Policy Self-Regularization

**Input:** training batch $[(s_i, g_i)]$, $i = 1, \cdots, M$, policy $\pi$, perturbation number for each transition $N$, the maximum perturbation $\epsilon'$, MSR threshold $\beta$, MSR strength $\lambda$
MSR_losses = [ ]
**for** transition $(s_i, g_i)$ **in** training batch **do**
   this_transition_MSR_losses = [ ]
   **for** $j = 1$ **to** $N$ **do**
      sample a perturbation $\epsilon \sim (-\epsilon', \epsilon')$
      tmp_MSR_loss = $\max\{D_{\mathrm{KL}}[\pi(\cdot|s_i, g_i), \pi(\cdot|s_i, g_i + \epsilon)] - \beta, 0\}$
      this_transition_MSR_losses.append(tmp_MSR_loss)
   **end for**
   MSR_losses.append(this_transition_MSR_losses.mean())
**end for**
return $\lambda \cdot$ MSR_losses.mean()

## 4. Experiments

### 4.1. Settings

To evaluate the effectiveness of the proposed method, we conduct experiments on three tasks across two physics simulation engines. The tasks included: (1) Reach and (2) Push tasks on the Panda robotic arm (Gallouédec et al., 2021), which are two multi-goal tasks frequently used in academic research. We modify the Reach and Push tasks to increase their difficulty. For Reach, we expand the target range, reduce the error threshold for determining achievement, and employ the *joints* control mode which means a larger action space; for Push, we expand the target range, employ the *joints* control mode and sparse rewards. (3) The Velocity Vector Control (VVC) task on a fixed-wing aircraft (Gong et al., 2025a), which is a typical multi-goal long-horizon task. The long-horizon nature of the VVC task implies that the algorithm must overcome more challenging exploration difficulties. The specific configurations for the three tasks are detailed in Appendix B. Overall, in terms of task difficulty, the order is Reach < Push < VVC.

We evaluate the following three baseline algorithms: (1) GC-SAC, representing off-policy GCRL algorithms; (2) HER, another off-policy GCRL algorithm that incorporates the goal-relabeling technique; and (3) GC-PPO, representing on-policy GCRL algorithms. Furthermore, we extend these baselines with MSR: (4) MSR-GC-SAC; (5) MSR-HER; and (6) MSR-GC-PPO. Detailed descriptions of these algorithms can be found in Appendix C.

### 4.2. Main Results

Table 1 presents the performance of our method and the baselines across three tasks.

**MSR effectively reduces the discrepancy between poli-**

*Table 1.* Policy discrepancy, $D_{\mathrm{KL}}(\pi_{g+\epsilon}, \pi_g)$, return gap, $J(\pi_{g+\epsilon}) - J(\pi_g)$, and expect return, $J(\pi)$, for different algorithms. Italics indicates that these results should be interpreted with caution due to the baseline algorithm's performance limitations. The mean and variance are shown over 10 random seeds.

(a) Reach

| Algorithms | $D_{\mathrm{KL}}(\pi_{g+\epsilon}, \pi_g)$ | $J(\pi_{g+\epsilon}) - J(\pi_g)$ | $J(\pi)$ |
|---|---|---|---|
| GC-SAC | $0.51_{\pm 1.28}$ | $-0.89_{\pm 0.68}$ | $-12.03_{\pm 0.63}$ |
| MSR-GC-SAC | $0.25_{\pm 0.03}$ | $0.00_{\pm 0.00}$ | $-0.85_{\pm 0.02}$ |
| HER | $0.71_{\pm 0.25}$ | $-0.01_{\pm 0.01}$ | $-1.16_{\pm 0.10}$ |
| MSR-HER | $0.31_{\pm 0.04}$ | $-0.01_{\pm 0.01}$ | $-0.96_{\pm 0.08}$ |
| GC-PPO | *$0.01_{\pm 0.03}$* | *$-5.29_{\pm 2.11}$* | *$-24.89_{\pm 1.62}$* |
| MSR-GC-PPO | $2.85_{\pm 0.72}$ | $-0.01_{\pm 0.02}$ | $-1.03_{\pm 0.15}$ |

(b) Push

| Algorithms | $D_{\mathrm{KL}}(\pi_{g+\epsilon}, \pi_g)$ | $J(\pi_{g+\epsilon}) - J(\pi_g)$ | $J(\pi)$ |
|---|---|---|---|
| GC-SAC | $9.69_{\pm 6.81}$ | $-7.58_{\pm 3.05}$ | $-25.50_{\pm 8.00}$ |
| MSR-GC-SAC | $4.27_{\pm 1.42}$ | $-6.48_{\pm 4.14}$ | $-23.80_{\pm 6.76}$ |
| HER | $5.62_{\pm 0.57}$ | $-7.05_{\pm 2.28}$ | $-18.00_{\pm 2.24}$ |
| MSR-HER | $3.80_{\pm 0.31}$ | $-7.02_{\pm 1.33}$ | $-16.31_{\pm 1.92}$ |
| GC-PPO | $0.80_{\pm 0.42}$ | $-8.21_{\pm 1.82}$ | $-24.76_{\pm 2.99}$ |
| MSR-GC-PPO | $0.61_{\pm 0.17}$ | $-7.08_{\pm 1.78}$ | $-23.08_{\pm 5.06}$ |

(c) VVC

| Algorithms | $D_{\mathrm{KL}}(\pi_{g+\epsilon}, \pi_g)$ | $J(\pi_{g+\epsilon}) - J(\pi_g)$ | $J(\pi)$ |
|---|---|---|---|
| GC-SAC | $0.37_{\pm 0.12}$ | $-22.00_{\pm 5.06}$ | $-138.20_{\pm 14.16}$ |
| MSR-GC-SAC | $0.33_{\pm 0.09}$ | $-20.94_{\pm 7.62}$ | $-132.09_{\pm 8.06}$ |
| HER | $0.65_{\pm 0.22}$ | $-7.91_{\pm 3.22}$ | $-69.21_{\pm 12.87}$ |
| MSR-HER | $0.58_{\pm 0.16}$ | $-7.52_{\pm 3.15}$ | $-64.73_{\pm 13.55}$ |
| GC-PPO | *$0.08_{\pm 0.08}$* | *$-44.50_{\pm 12.09}$* | *$-169.04_{\pm 25.57}$* |
| MSR-GC-PPO | $0.19_{\pm 0.20}$ | $-28.89_{\pm 14.75}$ | $-146.64_{\pm 28.05}$ |

**cies for adjacent desired goals, $D_{\mathrm{KL}}(\pi_{g+\epsilon}, \pi_g)$.** As observed in the second column of all tables, except for the comparison with GC-PPO on the Reach and VVC tasks, MSR consistently lowers $D_{\mathrm{KL}}(\pi_{g+\epsilon}, \pi_g)$ across all tasks and algorithms. By considering the policy return $J(\pi)$ and the success rate presented in Appendix D.2 and D.4, it is evident that GC-PPO barely acquires any capability to achieve goals on these two tasks. The failure to learn the distinctions between adjacent goals results in a small value of $D_{\mathrm{KL}}(\pi_{g+\epsilon}, \pi_g)$. Conversely, MSR aids the policy in learning the distinctions between adjacent goals, thereby slightly increasing $D_{\mathrm{KL}}(\pi_{g+\epsilon}, \pi_g)$.

**MSR also effectively diminishes the return gap between policies for adjacent desired goals, $|J(\pi_{g+\epsilon}) - J(\pi_g)|$.** The third column of all tables demonstrates that MSR reduces $|J(\pi_{g+\epsilon}) - J(\pi_g)|$ across all tasks and algorithms. In conjunction with Theorem 3.4 and the aforementioned analysis of $D_{\mathrm{KL}}(\pi_{g+\epsilon}, \pi_g)$, it can be concluded that MSR can effectively reduce the discrepancy between policies for adjacent desired goals, thereby lowering the return gap. Additionally, we present the success rate for adjacent desired goals surrounding achievable goals in Appendix D, which exhibits a similar trend to $|J(\pi_{g+\epsilon}) - J(\pi_g)|$.

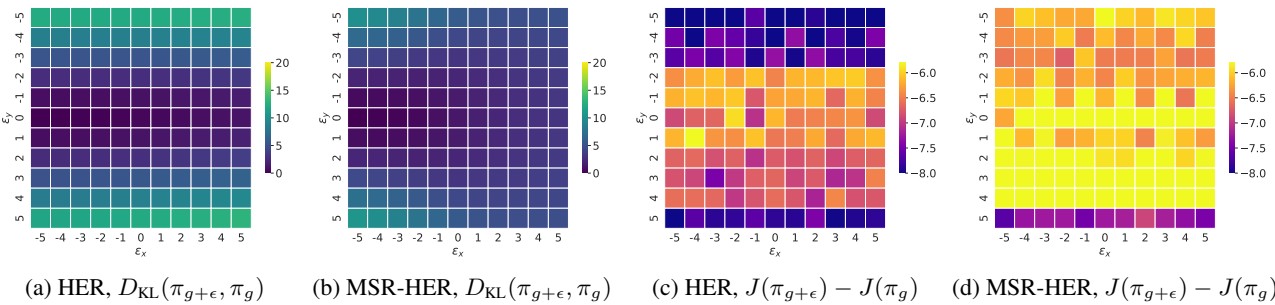

(a) HER, $D_{\mathrm{KL}}(\pi_{g+\epsilon}, \pi_g)$    (b) MSR-HER, $D_{\mathrm{KL}}(\pi_{g+\epsilon}, \pi_g)$    (c) HER, $J(\pi_{g+\epsilon}) - J(\pi_g)$    (d) MSR-HER, $J(\pi_{g+\epsilon}) - J(\pi_g)$

*Figure 3.* Policy discrepancy and return gap between policies for adjacent desired goals. Results come from experiments on Push over 10 ransom seeds. The interpretation of the coordinate axes and the data collection methods are analogous to those described in Fig. 1.

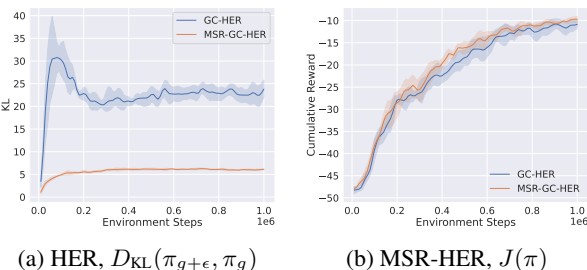

(a) HER, $D_{\mathrm{KL}}(\pi_{g+\epsilon}, \pi_g)$    (b) MSR-HER, $J(\pi)$

*Figure 4.* Trends of policy discrepancy between policies for adjacent desired goals and cumulative rewards of policy during training. It is important to note that the data presented in this figure are derived from the training process. In the computation of $D_{\mathrm{KL}}(\pi_{g+\epsilon}, \pi_g)$, $g$ may represent both achievable and unachievable goals. In contrast, the data shown in Fig. 3 are obtained from testing conducted after the completion of training, where $g$ is only comprised of achievable goals. Results come from HER and MSR-HER on Push over 10 random seeds.

**Furthermore, MSR enhances the cumulative rewards of the policy,** $J(\pi)$. The fourth column of all tables shows that MSR consistently improves $J(\pi)$. We posit that the enhancement of $J(\pi)$ by MSR is an indirect result of its reduction in $|J(\pi_{g+\epsilon}) - J(\pi_g)|$. Specifically, MSR's enhancement of the continuity in the policy's ability to achieve goals subsequently boosts the policy's capability to accumulate rewards across all goals.

Additionally, we analyze the impact of MSR on training efficiency. For detailed experiments and analysis, please refer to Appendix D.1.

In summary, the core capability of MSR is to reduce the policy discrepancy between adjacent desired goals. This reduction in policy discrepancy facilitates a decrease in the return gap between adjacent desired goals, thereby enhancing the overall ability of the policy to accumulate rewards.

### 4.3. A Micro-level Analysis of the Results

To provide a more intuitive demonstration of the effectiveness of our method, we present in Fig. 3 the policy dis-

crepancy and return gap between achievable goals and their adjacent goals. It can be observed that the color in Fig. 3b is darker than that in Fig. 3a, indicating that MSR-HER has a smaller $D_{\mathrm{KL}}(\pi_{g+\epsilon}, \pi_g)$ value, which suggests that our method effectively reduces the discrepancy between policies for adjacent desired goals. Furthermore, the color in Fig. 3d is lighter than that in Fig. 3c, indicating that the value of $J(\pi_{g+\epsilon}) - J(\pi_g)$ for MSR-HER is closer to zero, demonstrating that our method effectively reduces the return gap between policies for adjacent desired goals. For results on other tasks and algorithms, please refer to Appendix D.

In addition, Fig. 4 illustrates the trends of $D_{\mathrm{KL}}(\pi_{g+\epsilon}, \pi_g)$ and $J(\pi)$ during training. As observed in Fig. 4a, our approach successfully confines the values of $D_{\mathrm{KL}}(\pi_{g+\epsilon}, \pi_g)$ within a relatively low range, demonstrating its effectiveness in limiting the discrepancy between policies for adjacent desired goals without overly minimizing it. Furthermore, Fig. 4b reveals that the policy trained by our method exhibits a faster increase in cumulative rewards, indicating that our approach not only enhances the policy's capability to acquire cumulative rewards but also accelerates the rate at which these rewards are obtained.

### 4.4. Ablation Studies

In this section, we take the MSR-GC-SAC algorithm as an example on the Reach task to analyze the impact of the four hyperparameters in MSR, $\epsilon', \lambda, \beta, N$, on the training.

**Setting $\epsilon'$ to $\delta$ facilitates the reduction of discrepancy between policies for adjacent desired goals.**. The value of $\epsilon'$ determines the definition of adjacent desired goals, making it a parameter tightly coupled with the task. Typically, setting $\epsilon' = \delta$ is sufficient. To demonstrate the impact of $\epsilon'$ on training, we set $\lambda = 0.001, \beta = 0.0$ and $\epsilon'$ to $0.1\delta, 0.5\delta, \delta$, and respectively evaluate the trained policies within the ranges of $0.1\delta, 0.5\delta, \delta$ around achievable goals. Figure 5 presents the corresponding average $D_{\mathrm{KL}}(\pi_{g+\epsilon}, \pi_g)$ of achievable goals and their adjacent goals with $\epsilon$ sampled randomly from the range of evaluation $\epsilon'$. It can be observed that regardless of the value of $\epsilon'$ used in evaluation,

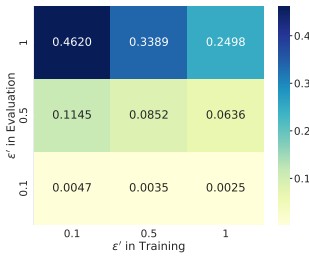

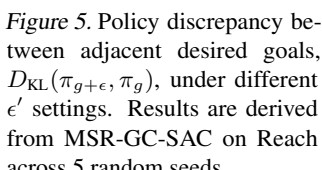

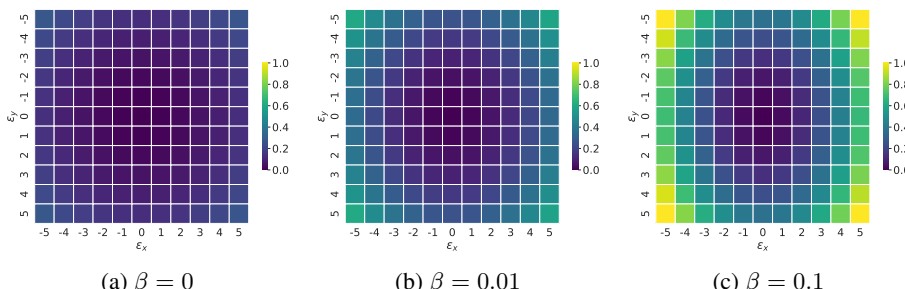

(a) $\beta = 0$      (b) $\beta = 0.01$      (c) $\beta = 0.1$

*Figure 5.* Policy discrepancy between adjacent desired goals, $D_{\mathrm{KL}}(\pi_{g+\epsilon}, \pi_g)$, under different $\epsilon'$ settings. Results are derived from MSR-GC-SAC on Reach across 5 random seeds.

*Figure 6.* Policy discrepancy between adjacent desired goals, $D_{\mathrm{KL}}(\pi_{g+\epsilon}, \pi_g)$, under different $\beta$ settings. The meanings of the coordinate axes and the evaluation methods are consistent with those in Fig.1. Results are derived from MSR-GC-SAC on Reach across 5 random seeds.

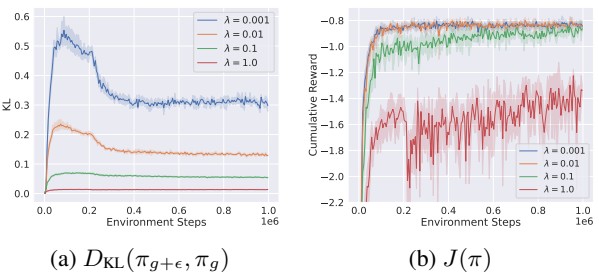

(a) $D_{\mathrm{KL}}(\pi_{g+\epsilon}, \pi_g)$      (b) $J(\pi)$

*Figure 7.* Trends of policy discrepancy between policies for adjacent desired goals, $D_{\mathrm{KL}}(\pi_{g+\epsilon}, \pi_g)$, and expected return, $J(\pi)$, during training under different $\lambda$ settings. Results are derived from MSR-GC-SAC on Reach across 5 random seeds.

*Table 2.* Policy discrepancy between adjacent desired goals, $D_{\mathrm{KL}}(\pi_{g+\epsilon}, \pi_g)$, return gap, $J(\pi_{g+\epsilon}) - J(\pi_g)$, and expected return, $J(\pi)$, under different $N$ settings. The mean and variance are shown over 5 random seeds.

| $N$ | $D_{\mathrm{KL}}(\pi_{g+\epsilon}, \pi_g)$ | $J(\pi_{g+\epsilon}) - J(\pi_g)$ | $J(\pi)$ |
|---|---|---|---|
| 1 | 0.23±0.18 | 0.00±0.01 | -0.85±0.01 |
| 4 | 0.24±0.19 | 0.00±0.01 | -0.83±0.02 |
| 16 | 0.25±0.20 | 0.00±0.01 | -0.84±0.02 |
| 64 | 0.22±0.17 | 0.00±0.01 | -0.82±0.01 |

increasing $\epsilon'$ used in training towards $\delta$ aids in reducing the discrepancy between policies for adjacent desired goals.

**The selection of $\lambda$ necessitates a balance between the effects of RL optimization and MSR optimization**. $\lambda$ is utilized to balance the relative importance between optimization objectives of RL and MSR. A larger value of $\lambda$ indicates a greater proportion of the MSR optimization objective in the overall optimization objective. We conduct ablation experiments with $\epsilon' = \delta, \beta = 0.1$ and $\lambda$ set at 0.001, 0.01, 0.1, and 1, and the trends of $D_{\mathrm{KL}}(\pi_{g+\epsilon}, \pi_g)$ and $J(\pi)$ during training are shown in Fig. 7. As observed in Fig. 7a, an increase in $\lambda$ corresponds to a decrease in the value of $D_{\mathrm{KL}}(\pi_{g+\epsilon}, \pi_g)$, suggesting that a larger $\lambda$ enhances the self-regularization effect on the policy. However, Fig. 7b reveals that a larger $\lambda$ results in a reduced capability of the policy to acquire cumulative rewards. This is attributed to the extension of the RL optimization objective by MSR; a greater emphasis on MSR leads to a diminished focus on the original RL optimization objective, adversely affecting the policy's ability to accumulate rewards. Therefore, $\lambda$ is a parameter that requires careful consideration when applying MSR. A too small $\lambda$ may not adequately leverage MSR's self-regularization effects on the policy, while an

excessively large $\lambda$ can compromise the optimization of the original RL objectives.

**A moderate increase in $\beta$ is beneficial for preventing MSR from over-optimizing the discrepancy between policies for adjacent desired goals**. The initial motivation for introducing the $\beta$ parameter into MSR is to prevent the over-optimization of the discrepancy between policies for adjacent desired goals during the policy optimization process. To validate the role of $\beta$, we conduct evaluations under the conditions $\epsilon' = 0.1\delta, \lambda = 1$ with $\beta$ set to 0.0, 0.01, and 0.1, and present the $D_{\mathrm{KL}}(\pi_{g+\epsilon}, \pi_g)$ between policies for adjacent desired goals in Fig. 6. It can be observed that as $\beta$ increases, the policy discrepancy also increases, effectively demonstrating $\beta$'s role in preventing the over-optimization of the discrepancy between policies for adjacent desired goals. However, it is important to note that an excessively large $\beta$ can diminish MSR's self-regularization effects on the policy during training, as an overly large $\beta$ implies that MSR is unable to function.

**MSR exhibits robustness to the value of $N$.** The role of $N$ is to average Equation 6 over multiple noises; as $N$ increases, the estimation of Equation 6 becomes more accurate, albeit with a corresponding increase in computational cost. To analyze the varying effects of different $N$ values on training, we conduct experiments with $N$ set to 1, 4, 16, and 64, under the conditions $\epsilon' = \delta, \lambda = 0.001, \beta = 0.1$.

Figure 8 illustrates the trend of $D_{\mathrm{KL}}(\pi_{g+\epsilon}, \pi_g)$ during training, while Table 2 presents the evaluation results after training. It can be observed that, both during training and in the final evaluation, there is negligible differential impact of varying $N$ values on the training process. This indicates that MSR is robust to the choice of $N$.

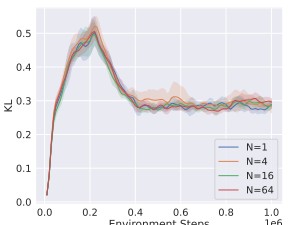

*Figure 8.* Policy discrepancy between adjacent desired goals, $D_{\mathrm{KL}}(\pi_{g+\epsilon}, \pi_g)$ under different $N$ settings. Results come from MSR-GC-SAC on Reach over 5 random seeds.

## 5. Related Work

**Goal-Conditioned Reinforcement Learning**. Research in GCRL primarily focuses on two directions. The first direction investigates sampling methods for behavior goals (Liu et al., 2022), which leverage the concept of curriculum learning (Narvekar et al., 2020) to select behavior goals from easy to difficult, thereby accelerating the convergence of the policy. This line of research encompasses two main questions: how to estimate the policy's ability to achieve goals, and how to select appropriate behavior goals based on this ability. In the general off-policy GCRL framework proposed by (Pitis et al., 2020), the policy's goal-achievement ability is estimated using data from the experience replay buffer. Conversely, in the general on-policy GCRL framework proposed by (Gong et al., 2024a), the policy's ability is assessed periodically during training, and the Off-Policy Evaluation (OPE) (Uehara et al., 2022) method is used to fit this ability based on historical evaluation data. Regarding methods for sampling behavior goals, approaches such as RIG (Nair et al., 2018), DISCERN (Warde-Farley et al., 2018), and MEGA (Pitis et al., 2020) sample behavior goals based on variations in the distribution of achieved goals. The second research direction explores sub-goal generation methods, which are primarily aimed at addressing the long-horizon challenge. These methods decompose the desired goal into several sub-goals (Park et al., 2024; Chane-Sane et al., 2021), and then use GCRL to sequentially solve these sub-goals, thereby achieving the original desired goal. It is evident that the primary research areas in GCRL have not yet taken note of the continuity of the policy's goal-achievement ability. To the best of our knowledge, our work is the first in the field of GCRL to focus on the continuity of the policy's goal-achievement ability.

**Generalization in Reinforcement Learning**. Our work shares similarities with research on generalization in RL (Korkmaz, 2024). For instance, (Lee et al., 2020) observed that RL agents frequently overfit to training environments and struggle to generalize to unseen environments. Simi-

larly, (Korkmaz, 2024) identified overfitting as a primary factor limiting the generalization capabilities of policies. Consequently, research on RL generalization has largely focused on overcoming overfitting. Examples include studies like (Laskin et al., 2020; Yarats et al., 2021), which concentrate on data regularization through data augmentation methods, and (Igl et al., 2019; Lee et al., 2020; Liu et al., 2021), which focus on policy regularization via neural network regularization techniques. In the context of GCRL, (Yang et al., 2023) combined both data and policy regularization methods to enhance the generalization ability of offline GCRL approaches. Despite these similarities, our research differs significantly from generalization studies. We discuss the continuity of the policy's goal-achievement ability, without emphasizing whether it pertains to training or unseen environments. As our experiments demonstrate, for tasks with continuous state spaces, the policy's continuity of goal-achievement ability is suboptimal even within the training tasks.

Additionally, KL divergence is commonly used as a measure of distance to constrain policies in policy optimization. In the context of multi-goal settings, we compare MSR with two KL-based policy constraint approaches and provide a detailed analysis in Appendix E.

## 6. Conclusion and Limitations

In this paper, we delve into the challenge of discontinuity in goal-achievement capabilities inherent in Goal-conditioned Reinforcement Learning (GCRL) algorithms. Through a comprehensive evaluation of various prevalent GCRL algorithms across diverse tasks, we empirically demonstrate the prevalence and significance of this issue. Our theoretical analysis provide deeper insights into the underlying causes of this discontinuity, highlighting the importance of regularizing policy discrepancies between achieved goals and their adjacent goals.

In response to these findings, we propose a novel margin-based policy self-regularization (MSR) approach. MSR, designed to optimize the minimal acceptable discrepancy between policies for a desired goal and its adjacent goals, offers a robust solution to the discontinuity problem. By integrating MSR as a plug-in module into existing GCRL algorithms, we show its effectiveness through systematic evaluations on two robotic arm control tasks and a complex, real-world fixed-wing aircraft control task.

Some limitations should be addressed in future work. Firstly, exploring the scalability of our method to even more complex tasks and environments, as well as investigating its applicability to more GCRL algorithms, remain important directions. Secondly, further theoretical analysis could provide more nuanced understandings of the relation-

ship between policy differences and cumulative rewards. Thirdly, our approach mitigates the discontinuity in goal-achievement capabilities under expected scenarios, but it does not address the discontinuity in goal-achievement capabilities in the worst-case scenarios.

In conclusion, our research contributes to the advancement of GCRL by providing a novel and effective solution to the discontinuity problem, thereby enhancing the performance and applicability of GCRL algorithms in real-world scenarios.

## Acknowledgements

This work was supported by the Open Fund of National Key Laboratory of Parallel and Distributed Computing (PDL) (NO.2024-KJWPDL-02) and the Science and Technology Innovation Program of Hunan Province (No.2023RC1005).

## Impact Statement

This paper presents work whose goal is to advance the field of Machine Learning. There are many potential societal consequences of our work, none which we feel must be specifically highlighted here.

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

# Appendix

## A. Theoretical Proofs

### A.1. Useful Lemmas

**Lemma A.1.** *(Pinsker's inequality (Csiszár & Körner, 2011)) For two arbitrary distributions $\mu$ and $\nu$, $D_{TV}(\mu, \nu) \leq \sqrt{2D_{KL}(\mu, \nu)}$*

**Lemma A.2.** *(Xu et al., 2020) For any two policies $\pi_1$ and $\pi_2$, we have that*

$$D_{TV}(\rho_{\pi_1}, \rho_{\pi_2}) \leq \frac{1}{1-\gamma}\mathbb{E}_{s \sim d_{\pi_1}} \left[D_{TV}\left(\pi_1(\cdot|s), \pi_2(\cdot|s)\right)\right]. \tag{8}$$

### A.2. Proof of Theorem 3.1

*Proof.* Based on that $\phi(s_T) = g$ and $\|\epsilon\| < \delta$, we can derive that $\|\phi(s_T) - (g + \epsilon)\| = \|g - (g + \epsilon)\| = \|\epsilon\| \leq \delta$. $\qquad\square$

### A.3. Proof of Corollary 3.2

*Proof.* We give the proof under the conditions of deterministic environment transitions and deterministic policies.

Suppose that the trajectory $\tau$ achieved by policy $\pi$ for goal $g'$ is $\tau = (s_0, a_0, s_1, a_1, \cdots, s_T, a_T)$, where $a_i = \pi(s_i, g'), 1 \leq i \leq T$, and it satisfies $\phi(s_T) = g$ with $\|g' - g\| < \delta$. For the goal $g + \epsilon$, let the corresponding trajectory be denoted as $\tau' = (s'_0, a'_0, s'_1, a'_1, \cdots, s'_T, a'_T)$, where $a'_i = \pi(s'_i, g + \epsilon), 1 \leq i \leq T$.

We proceed to prove that $\tau = \tau'$ by induction:

Firstly, since we assume a fixed initial state, it follows that $s_0 = s'_0$. Furthermore, from the condition $\mathbb{E}_{s \sim d_{\pi_{g'}}} D_{KL}\left(\pi(s, g'), \pi(s, g + \epsilon)\right) = 0$, we can infer that $a'_0 = \pi(s'_0, g + \epsilon) = \pi(s_0, g + \epsilon) = \pi(s_0, g') = a_0$, which implies $\tau_{0:1} = \tau'_{0:1}$.

Secondly, assuming $\tau_{0:k} = \tau'_{0:k}$, we have $s_k = \mathcal{T}(s_{k-1}, a_{k-1}) = \mathcal{T}(s'_{k-1}, a'_{k-1}) = s'_k$, and $a_k = \pi(s_k, g') = \pi(s_k, g + \epsilon) = \pi(s'_k, g + \epsilon) = a'_k$, thus $\tau_{0:k+1} = \tau'_{0:k+1}$.

The above proof establishes that $\pi(\cdot, g + \epsilon)$ can yield the same trajectory $\tau$ as $\pi(\cdot, g')$. By combining this with Theorem 3.1, we conclude the proof.

$\qquad\square$

### A.4. Proof of Theorem 3.3

*Proof.*

$$J(\pi_{g+\epsilon}) \tag{9}$$

$$= \frac{1}{1-\gamma} \sum_{(s,a)} \rho_{\pi_{g+\epsilon}}(s,a) r(s,a,g+\epsilon) \tag{10}$$

$$= \frac{1}{1-\gamma} \sum_{(s,a)} (\rho_{\pi_{g+\epsilon}}(s,a) - \rho_{\pi_g}(s,a)) r(s,a,g+\epsilon) + \frac{1}{1-\gamma} \sum_{(s,a)} \rho_{\pi_g}(s,a) r(s,a,g+\epsilon) \tag{11}$$

$$\geq -\frac{1}{1-\gamma} \sum_{(s,a)} \left|\rho_{\pi_{g+\epsilon}}(s,a) - \rho_{\pi_g}(s,a)\right| r(s,a,g+\epsilon) + \frac{1}{1-\gamma} \sum_{(s,a)} \rho_{\pi_g}(s,a) r(s,a,g+\epsilon) \tag{12}$$

$$\geq -\frac{R_{\max}}{1-\gamma} \sum_{(s,a)} \left|\rho_{\pi_{g+\epsilon}}(s,a) - \rho_{\pi_g}(s,a)\right| + \frac{1}{1-\gamma} \sum_{(s,a)} \rho_{\pi_g}(s,a) r(s,a,g+\epsilon) \tag{13}$$

$$= -\frac{2R_{\max}}{1-\gamma} D_{\text{TV}}(\rho_{\pi_{g+\epsilon}}, \rho_{\pi_g}) + \frac{1}{1-\gamma} \sum_{(s,a)} \rho_{\pi_g}(s,a)r(s,a,g+\epsilon) \tag{14}$$

$$\geq -\frac{2R_{\max}}{(1-\gamma)^2} \mathbb{E}_{s \sim d_{\pi_g}} \left[ D_{\text{TV}} \left( \pi_{g+\epsilon}(\cdot|s), \pi_g(\cdot|s) \right) \right] + \frac{1}{1-\gamma} \sum_{(s,a)} \rho_{\pi_g}(s,a)r(s,a,g+\epsilon) \tag{15}$$

$$\geq -\frac{2\sqrt{2}R_{\max}}{(1-\gamma)^2} \sqrt{D_{\text{KL}} \left( \pi_{g+\epsilon}(\cdot|s), \pi_g(\cdot|s) \right)} + \frac{1}{1-\gamma} \sum_{(s,a)} \rho_{\pi_g}(s,a)r(s,a,g+\epsilon) \tag{16}$$

$$J(\pi_{g+\epsilon}) \tag{17}$$

$$= \frac{1}{1-\gamma} \sum_{(s,a)} \rho_{\pi_{g+\epsilon}}(s,a)r(s,a,g+\epsilon) \tag{18}$$

$$= \frac{1}{1-\gamma} \sum_{(s,a)} (\rho_{\pi_{g+\epsilon}}(s,a) - \rho_{\pi_g}(s,a))r(s,a,g+\epsilon) + \frac{1}{1-\gamma} \sum_{(s,a)} \rho_{\pi_g}(s,a)r(s,a,g+\epsilon) \tag{19}$$

$$\leq \frac{1}{1-\gamma} \sum_{(s,a)} \left| \rho_{\pi_{g+\epsilon}}(s,a) - \rho_{\pi_g}(s,a) \right| r(s,a,g+\epsilon) + \frac{1}{1-\gamma} \sum_{(s,a)} \rho_{\pi_g}(s,a)r(s,a,g+\epsilon) \tag{20}$$

$$\leq \frac{R_{\max}}{1-\gamma} \sum_{(s,a)} \left| \rho_{\pi_{g+\epsilon}}(s,a) - \rho_{\pi_g}(s,a) \right| + \frac{1}{1-\gamma} \sum_{(s,a)} \rho_{\pi_g}(s,a)r(s,a,g+\epsilon) \tag{21}$$

$$= \frac{2R_{\max}}{1-\gamma} D_{\text{TV}}(\rho_{\pi_{g+\epsilon}}, \rho_{\pi_g}) + \frac{1}{1-\gamma} \sum_{(s,a)} \rho_{\pi_g}(s,a)r(s,a,g+\epsilon) \tag{22}$$

$$\leq \frac{2R_{\max}}{(1-\gamma)^2} \mathbb{E}_{s \sim d_{\pi_g}} \left[ D_{\text{TV}} \left( \pi_{g+\epsilon}(\cdot|s), \pi_g(\cdot|s) \right) \right] + \frac{1}{1-\gamma} \sum_{(s,a)} \rho_{\pi_g}(s,a)r(s,a,g+\epsilon) \tag{23}$$

$$\leq \frac{2\sqrt{2}R_{\max}}{(1-\gamma)^2} \sqrt{D_{\text{KL}} \left( \pi_{g+\epsilon}(\cdot|s), \pi_g(\cdot|s) \right)} + \frac{1}{1-\gamma} \sum_{(s,a)} \rho_{\pi_g}(s,a)r(s,a,g+\epsilon) \tag{24}$$

□

### A.5. Proof of Theorem 3.4

*Proof.*

$$|J(\pi_{g_1}) - J(\pi_{g_2})| \tag{25}$$

$$= \frac{1}{1-\gamma} \left| \mathbb{E}_{(s,a) \sim \rho_{\pi_{g_1}}} r(s,a,g_1) - \mathbb{E}_{(s,a) \sim \rho_{\pi_{g_2}}} r(s,a,g_2) \right| \tag{26}$$

$$= \frac{1}{1-\gamma} \sum_{(s,a)} \left| \rho_{\pi_{g_1}}(s,a)r(s,a,g_1) - \rho_{\pi_{g_2}}(s,a)r(s,a,g_2) \right| \tag{27}$$

$$= \frac{1}{1-\gamma} \sum_{(s,a)} \left| \rho_{\pi_{g_1}}(s,a)r(s,a,g_1) - \rho_{\pi_{g_1}}(s,a)r(s,a,g_2) + \rho_{\pi_{g_1}}(s,a)r(s,a,g_2) - \rho_{\pi_{g_2}}(s,a)r(s,a,g_2) \right| \tag{28}$$

$$\leq \frac{1}{1-\gamma} \sum_{(s,a)} \left| \rho_{\pi_{g_1}}(s,a) \left[ r(s,a,g_1) - r(s,a,g_2) \right] \right| + \frac{1}{1-\gamma} \sum_{(s,a)} \left| r(s,a,g_2) \left[ \rho_{\pi_{g_1}}(s,a) - \rho_{\pi_{g_2}}(s,a) \right] \right| \tag{29}$$

$$\leq \frac{1}{1-\gamma} \sum_{(s,a)} \left| \rho_{\pi_{g_1}}(s,a) \left[ r(s,a,g_1) - r(s,a,g_2) \right] \right| + \frac{R_{\max}}{1-\gamma} \sum_{(s,a)} \left| \left[ \rho_{\pi_{g_1}}(s,a) - \rho_{\pi_{g_2}}(s,a) \right] \right| \tag{30}$$

$$= \frac{1}{1-\gamma} \sum_{(s,a)} \rho_{\pi_{g_1}}(s,a) \left| r(s,a,g_1) - r(s,a,g_2) \right| + \frac{2R_{\max}}{1-\gamma} D_{\text{TV}} \left( \rho_{\pi_{g_1}}, \rho_{\pi_{g_2}} \right) \tag{31}$$

*Table 3.* Environment Hyper-Parameters

| (a) Reach | | | (b) Push | | | (c) VVC | |
|---|---|---|---|---|---|---|---|
| Parameter | Value | | Parameter | Value | | Parameter | Value |
| control_type | joints | | control_type | joints | | v_min | 150 |
| reward_type | dense | | reward_type | sparse | | v_max | 250 |
| goal_range | 0.5 | | goal_xy_range | 0.5 | | mu_min | -30 |
| distance_threshold | 0.01 | | obj_xy_range | 0.0 | | mu_max | 30 |
| max_episode_steps | 50 | | distance_threshold | 0.05 | | chi_min | -60 |
| | | | max_episode_steps | 50 | | chi_max | 60 |
| | | | | | | max_episode_steps | 400 |

$$\leq \frac{1}{1-\gamma} \sum_{(s,a)} \rho_{\pi_{g_1}}(s,a) \left| r(s,a,g_1) - r(s,a,g_2) \right| + \frac{2\sqrt{2}R_{\max}}{(1-\gamma)^2} \mathbb{E}_{s \sim d_{\pi_{g_1}}} \sqrt{d_{\mathrm{KL}}(\pi_{g_1}(\cdot|s), \pi_{g_2}(\cdot|s))} \tag{32}$$

$\square$

## A.6. Proof of Corollary 3.5

*Proof.*

$$\left| J(\pi_{g_1}) - J(\pi_{g_2}) \right| \tag{33}$$

$$\leq \frac{1}{1-\gamma} \sum_{(s,a)} \rho_{\pi_{g_1}}(s,a) \left| r(s,a,g_1) - r(s,a,g_2) \right| + \frac{2\sqrt{2}R_{\max}}{(1-\gamma)^2} \mathbb{E}_{s \sim d_{\pi_{g_1}}} \sqrt{d_{\mathrm{KL}}(\pi_{g_1}(\cdot|s), \pi_{g_2}(\cdot|s))} \tag{34}$$

$$= \frac{1}{1-\gamma} \sum_{(s,a)} \rho_{\pi_{g_1}}(s,a) \left| -\|\phi(s) - g_1\| + \|\phi(s) - g_2\| \right| + \frac{2\sqrt{2}R_{\max}}{(1-\gamma)^2} \mathbb{E}_{s \sim d_{\pi_{g_1}}} \sqrt{d_{\mathrm{KL}}(\pi_{g_1}(\cdot|s), \pi_{g_2}(\cdot|s))} \tag{35}$$

$$\leq \frac{1}{1-\gamma} \sum_{(s,a)} \rho_{\pi_{g_1}}(s,a) \|g_1 - g_2\| + \frac{2\sqrt{2}R_{\max}}{(1-\gamma)^2} \mathbb{E}_{s \sim d_{\pi_{g_1}}} \sqrt{d_{\mathrm{KL}}(\pi_{g_1}(\cdot|s), \pi_{g_2}(\cdot|s))} \tag{36}$$

$$= \frac{\|g_1 - g_2\|}{1-\gamma} + \frac{2\sqrt{2}R_{\max}}{(1-\gamma)^2} \mathbb{E}_{s \sim d_{\pi_{g_1}}} \sqrt{d_{\mathrm{KL}}(\pi_{g_1}(\cdot|s), \pi_{g_2}(\cdot|s))} \tag{37}$$

$\square$

## A.7. Proof of Corollary 3.6

*Proof.*

$$\left| J(\pi_g) - J(\pi_{g+\epsilon}) \right| \tag{38}$$

$$\leq \frac{\|g - (g+\epsilon)\|}{1-\gamma} + \frac{2\sqrt{2}R_{\max}}{(1-\gamma)^2} \mathbb{E}_{s \sim d_{\pi_g}} \sqrt{d_{\mathrm{KL}}(\pi_g(\cdot|s), \pi_{g+\epsilon}(\cdot|s))} \tag{39}$$

$$\leq \frac{\delta}{1-\gamma} + \frac{2\sqrt{2}R_{\max}}{(1-\gamma)^2} \mathbb{E}_{s \sim d_{\pi_g}} \sqrt{d_{\mathrm{KL}}(\pi_g(\cdot|s), \pi_{g+\epsilon}(\cdot|s))} \tag{40}$$

$\square$

# B. Environments Details

The configurations for the Reach and Push tasks are presented in Tables 3a and 3b, respectively, with all other parameters adhering to the default settings of Panda-Gym (Panda-Gym, 2024). For the VVC task, the settings are detailed in Table 3c, while the remaining parameters are maintained at the default settings of FlyCraft (Gong et al., 2025a).

*Table 4.* Algorithm Hyper-Parameters Used on Reach and Push

| (a) SAC | | (b) HER | | (c) PPO | |
|---|---|---|---|---|---|
| Parameter | Value | Parameter | Value | Parameter | Value |
| net_arch | [256,256] | net_arch | [256,256] | net_arch | [256,256] |
| gamma | 0.95 | gamma | 0.95 | gamma | 0.95 |
| train_steps | $1 \times 10^6$ | train_steps (Reach/Push) | $5 \times 10^4/1 \times 10^6$ | train_steps | $1 \times 10^7$ |
| batch_size | 256 | batch_size | 256 | batch_size | 512 |
| rollout_process_num | 1 | rollout_process_num | 1 | n_steps | 512 |
| learning_rate | $3 \times 10^{-4}$ | learning_rate | $3 \times 10^{-4}$ | n_epochs | 5 |
| gradient_steps | 1 | gradient_steps | 1 | rollout_process_num | 16 |
| buffer_size | $2 \times 10^5$ | buffer_size | $2 \times 10^5$ | learning_rate | $3 \times 10^{-4}$ |

*Table 5.* Algorithm Hyper-Parameters Used on VVC

| (a) SAC | | (b) HER | | (c) PPO | |
|---|---|---|---|---|---|
| Parameter | Value | Parameter | Value | Parameter | Value |
| net_arch | [128,128] | net_arch | [128,128] | net_arch | [128,128] |
| gamma | 0.995 | gamma | 0.995 | gamma | 0.95 |
| train_steps | $10^6$ | train_steps | $5 * 10^5$ | train_steps | $2 \times 10^8$ |
| batch_size | 1024 | batch_size | 1024 | batch_size | 4096 |
| rollout_process_num | 1 | rollout_process_num | 1 | n_steps | 512 |
| learning_rate | $3 \times 10^{-4}$ | learning_rate | $3 \times 10^{-4}$ | n_epochs | 5 |
| gradient_steps | 1 | gradient_steps | 1 | rollout_process_num | 64 |
| buffer_size | $2 \times 10^5$ | buffer_size | $2 \times 10^5$ | learning_rate | $3 \times 10^{-4}$ |

*Table 6.* Comparison of training wall clock time between baselines and MSR+baselines. Results come from PPO ($10^7$ environment steps), SAC ($10^5$ environment steps), and HER ($10^5$ environment steps) on VVC.

| | GC-PPO | MSR-GC-PPO | GC-SAC | MSR-GC-SAC | HER | MSR-HER |
|---|---|---|---|---|---|---|
| Wall Clock Time (s) | 3498 | 3573 | 2621 | 3175 | 5096 | 5912 |

## C. Implementation Details

All our GC-SAC, GC-HER, and GC-PPO are implemented with the Stable-Baselines3 (Raffin et al., 2021) framework. For our MSR method, we employ $\epsilon' = \delta, \lambda = 10^{-3}, \beta = 0.1, N = 16$ for Reach and Push and $\epsilon' = 0.1 \cdot \delta, \lambda = 10^{-3}, \beta = 0.1, N = 16$ for VVC.

## D. Additional Results

In this section, we provide supplementary presentations of the experimental results discussed in the main text.

### D.1. Impact on Training Efficiency

MSR adds training overhead without influencing inference, due to added regularization only in policy optimization. Table 6 shows the wall clock time for training different baselines and baselines+MSR. For on-policy RL, where the main cost is environment sampling, MSR's effect is negligible, with only a 2.14% increase. In off-policy RL, which leverages replay buffer data, the impact is more pronounced: a 21.14% increase for MSR-GC-SAC and 16.02% for MSR-HER. Figs. 13, 17, and 22 reveal MSR's convergence rate is on par with baselines. Overall, despite the slight added training overhead, MSR markedly improves policy continuity in achieving goals.

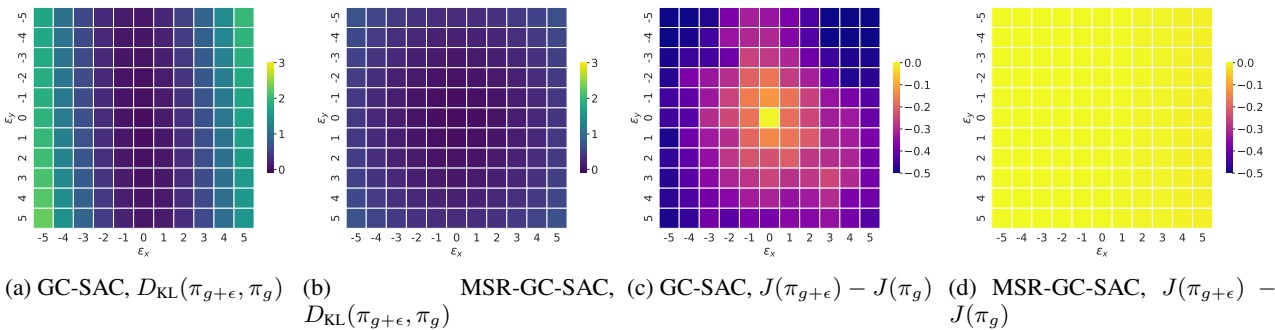

(a) GC-SAC, $D_{\mathrm{KL}}(\pi_{g+\epsilon}, \pi_g)$   (b) MSR-GC-SAC, $D_{\mathrm{KL}}(\pi_{g+\epsilon}, \pi_g)$   (c) GC-SAC, $J(\pi_{g+\epsilon}) - J(\pi_g)$   (d) MSR-GC-SAC, $J(\pi_{g+\epsilon}) - J(\pi_g)$

*Figure 9.* Policy discrepancy and return gap between policies for adjacent desired goals. Results come from experiments on **Reach** with **GC-SAC** and **MSR-GC-SAC**. The interpretation of the coordinate axes and the data collection methods are analogous to those described in Fig. 1.

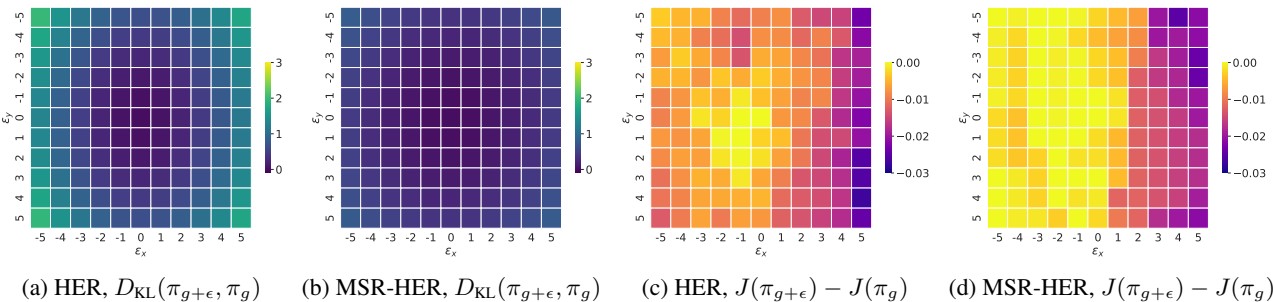

(a) HER, $D_{\mathrm{KL}}(\pi_{g+\epsilon}, \pi_g)$   (b) MSR-HER, $D_{\mathrm{KL}}(\pi_{g+\epsilon}, \pi_g)$   (c) HER, $J(\pi_{g+\epsilon}) - J(\pi_g)$   (d) MSR-HER, $J(\pi_{g+\epsilon}) - J(\pi_g)$

*Figure 10.* Policy discrepancy and return gap between policies for adjacent desired goals. Results come from experiments on **Reach** with **HER** and **MSR-HER**. The interpretation of the coordinate axes and the data collection methods are analogous to those described in Fig. 1.

### D.2. On Reach

Figure 9 illustrates the evaluation results of GC-SAC and MSR-GC-SAC. It is evident that MSR-GC-SAC significantly reduces the policy discrepancy between adjacent desired goals, $D_{\mathrm{KL}}(\pi_{g+\epsilon}, \pi_g)$, while effectively diminishing the return gap between policies for adjacent desired goals, $J(\pi_{g+\epsilon}) - J(\pi_g)$. Figure 10 presents the evaluation results of HER and MSR-HER, yielding conclusions similar to those of MSR-GC-SAC. Figure 11 depicts the evaluation results of GC-PPO and MSR-GC-PPO, showing that MSR-GC-PPO slightly increases $J(\pi_{g+\epsilon}) - J(\pi_g)$. In conjunction with the success rates presented in Table 7, it is observed that the success rate of GC-PPO is nearly zero, suggesting that GC-PPO lacks the capability to achieve goals. This is reflected in $D_{\mathrm{KL}}(\pi_{g+\epsilon}, \pi_g)$, indicating that the policy cannot distinguish between different goals, resulting in a particularly small value of $D_{\mathrm{KL}}(\pi_{g+\epsilon}, \pi_g)$.

Figure 12 shows the trend of policy discrepancy between adjacent desired goals, $D_{\mathrm{KL}}(\pi_{g+\epsilon}, \pi_g)$, during training, indicating that MSR maintains $D_{\mathrm{KL}}(\pi_{g+\epsilon}, \pi_g)$ within a relatively small but not excessively small range. Figure 13 demonstrates the trend of cumulative rewards during training, revealing that MSR not only enhances the upper limit of cumulative rewards acquisition but also accelerates the rate of cumulative rewards acquisition.

Table 7 presents the success rates of adjacent goals around an achievable goal and the overall success rate. Given that our method diminishes the return gap between policies for adjacent desired goals and enhances the overall capability of the policy to acquire cumulative rewards, the policy exhibits a notable improvement in both the success rate of adjacent goals around an achievable goal and the overall success rate.

### D.3. On Push

Figure 14 illustrates the evaluation results of GC-SAC and MSR-GC-SAC. It is evident that MSR-GC-SAC significantly reduces the policy discrepancy between adjacent desired goals, $D_{\mathrm{KL}}(\pi_{g+\epsilon}, \pi_g)$, while effectively diminishing the return gap between policies for adjacent desired goals, $J(\pi_{g+\epsilon}) - J(\pi_g)$. Figure 15 presents the evaluation results of GC-PPO and MSR-GC-PPO, yielding conclusions similar to those of MSR-GC-SAC.

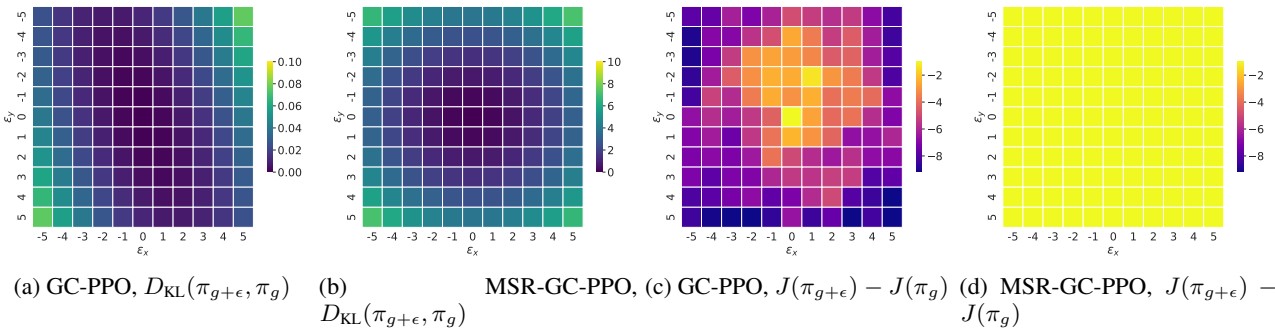

(a) GC-PPO, $D_{\mathrm{KL}}(\pi_{g+\epsilon}, \pi_g)$ (b) $D_{\mathrm{KL}}(\pi_{g+\epsilon}, \pi_g)$ MSR-GC-PPO, (c) GC-PPO, $J(\pi_{g+\epsilon}) - J(\pi_g)$ (d) MSR-GC-PPO, $J(\pi_{g+\epsilon}) - J(\pi_g)$

*Figure 11.* Policy discrepancy and return gap between policies for adjacent desired goals. Results come from experiments on **Reach** with **GC-PPO** and **MSR-GC-PPO**. The interpretation of the coordinate axes and the data collection methods are analogous to those described in Fig. 1.

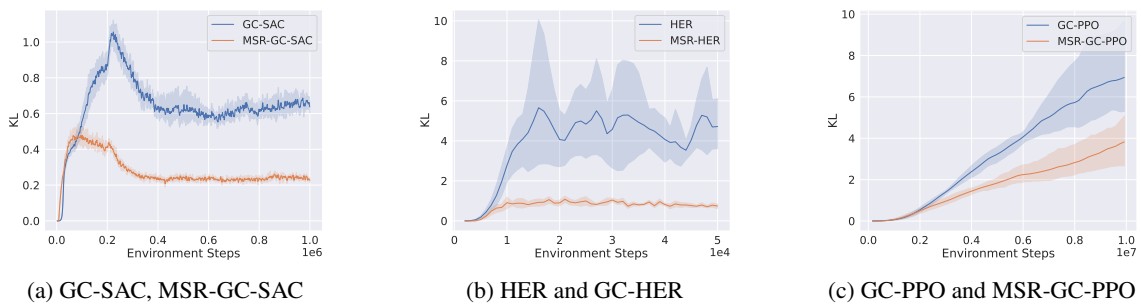

(a) GC-SAC, MSR-GC-SAC        (b) HER and GC-HER        (c) GC-PPO and MSR-GC-PPO

*Figure 12.* Trends of policy discrepancy between policies for adjacent desired goals during training on **Reach**. Results come from algorithms over 5 random seeds.

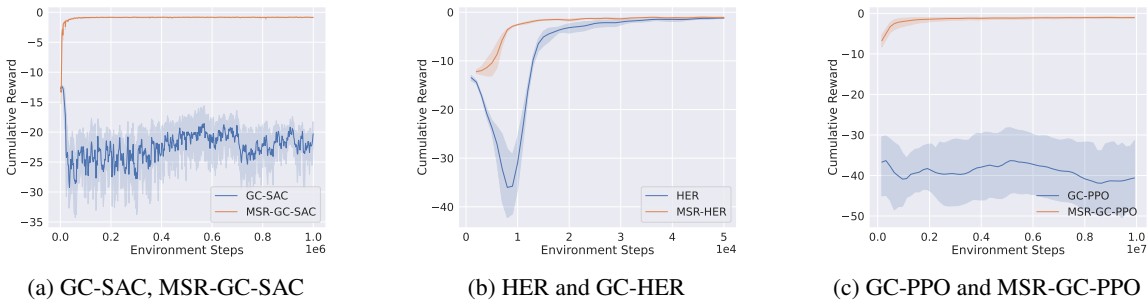

(a) GC-SAC, MSR-GC-SAC        (b) HER and GC-HER        (c) GC-PPO and MSR-GC-PPO

*Figure 13.* Cumulative rewards of policy during training on **Reach**. Results come from algorithms over 5 random seeds.

*Table 7.* Success rate of adjacent goals around an achievable goal and overall success rate of policy for different algorithms on **Reach**. The mean and variance are shown over 5 random seeds. Italics indicates that these results should be interpreted with caution due to the baseline algorithm's performance limitations.

| Algorithms | Adjacent Success Rate (%) | Success Rate (%) |
|---|---|---|
| GC-SAC | *65.58±9.10* | *0.20±0.29* |
| MSR-GC-SAC | 99.98±0.04 | 99.95±0.12 |
| HER | 98.98±0.42 | 94.60±4.43 |
| MSR-HER | 99.34±0.27 | 90.69±6.85 |
| GC-PPO | *44.63±13.83* | *0.04±0.09* |
| MSR-GC-PPO | 98.79±0.36 | 98.05±3.19 |

Figure 16 shows the trend of policy discrepancy between adjacent desired goals, $D_{\mathrm{KL}}(\pi_{g+\epsilon}, \pi_g)$, during training. Figure 17

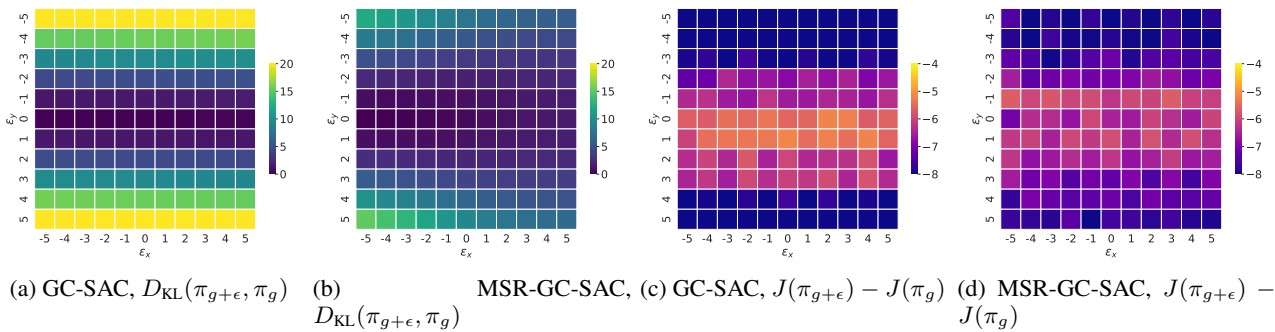

(a) GC-SAC, $D_{\mathrm{KL}}(\pi_{g+\epsilon}, \pi_g)$ (b) MSR-GC-SAC, (c) GC-SAC, $J(\pi_{g+\epsilon}) - J(\pi_g)$ (d) MSR-GC-SAC, $J(\pi_{g+\epsilon}) - J(\pi_g)$

*Figure 14.* Policy discrepancy and return gap between policies for adjacent desired goals. Results come from experiments on **Push** with **GC-SAC** and **MSR-GC-SAC**. The interpretation of the coordinate axes and the data collection methods are analogous to those described in Fig. 1.

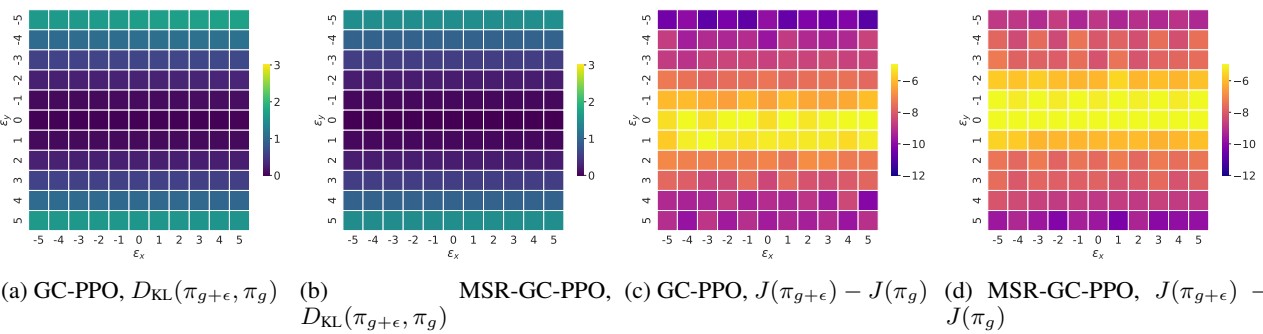

(a) GC-PPO, $D_{\mathrm{KL}}(\pi_{g+\epsilon}, \pi_g)$ (b) MSR-GC-PPO, (c) GC-PPO, $J(\pi_{g+\epsilon}) - J(\pi_g)$ (d) MSR-GC-PPO, $J(\pi_{g+\epsilon}) - J(\pi_g)$

*Figure 15.* Policy discrepancy and return gap between policies for adjacent desired goals. Results come from experiments on **Push** with **GC-PPO** and **MSR-GC-PPO**. The interpretation of the coordinate axes and the data collection methods are analogous to those described in Fig. 1.

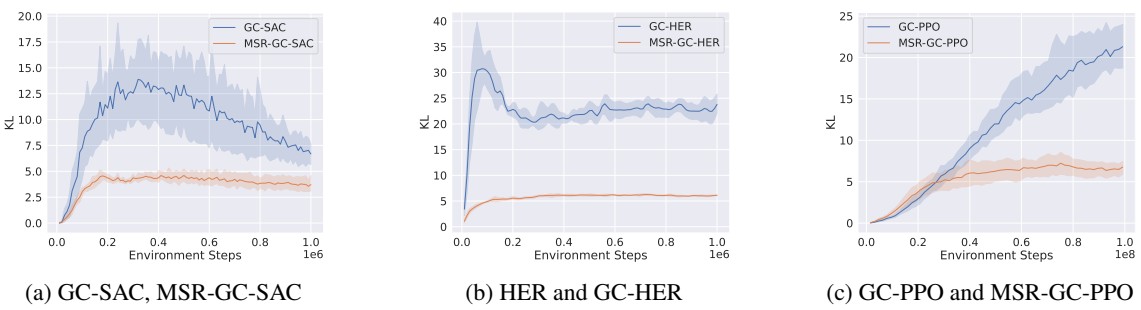

(a) GC-SAC, MSR-GC-SAC      (b) HER and GC-HER      (c) GC-PPO and MSR-GC-PPO

*Figure 16.* Trends of policy discrepancy between policies for adjacent desired goals during training on **Push**. Results come from algorithms over 5 random seeds.

demonstrates the trend of cumulative rewards during training. Table 8 presents the success rates of adjacent goals around an achievable goal and the overall success rate. All of the above results demonstrated conclusions similar to those observed on the Push task.

### D.4. On VVC

Figure 18 illustrates the evaluation results of GC-SAC and MSR-GC-SAC. Figure 19 illustrates the evaluation results of HER and MSR-HER. Figure 20 illustrates the evaluation results of GC-PPO and MSR-GC-PPO.

Figure 21 shows the trend of policy discrepancy between adjacent desired goals, $D_{\mathrm{KL}}(\pi_{g+\epsilon}, \pi_g)$, during training. Figure 22 demonstrates the trend of cumulative rewards during training. Table 9 presents the success rates of adjacent goals around an achievable goal and the overall success rate. All of the above results demonstrated conclusions similar to those observed on

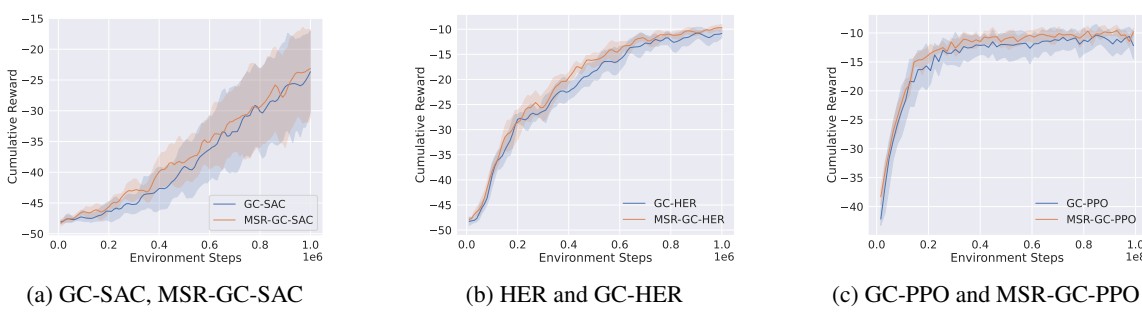

(a) GC-SAC, MSR-GC-SAC      (b) HER and GC-HER      (c) GC-PPO and MSR-GC-PPO

*Figure 17.* Cumulative rewards of policy during training on **Push**. Results come from algorithms over 5 random seeds.

*Table 8.* Success rate of adjacent goals around an achievable goal and overall success rate of policy for different algorithms on **Push**. The mean and variance are shown over 5 random seeds.

| Algorithms | Adjacent Success Rate (%) | Success Rate (%) |
|---|---|---|
| GC-SAC | $81.26_{\pm 3.28}$ | $58.37_{\pm 19.37}$ |
| MSR-GC-SAC | $84.37_{\pm 2.34}$ | $62.17_{\pm 16.23}$ |
| HER | $82.27_{\pm 1.34}$ | $79.49_{\pm 5.34}$ |
| MSR-HER | $82.29_{\pm 1.51}$ | $84.27_{\pm 3.92}$ |
| GC-PPO | $77.07_{\pm 3.36}$ | $66.90_{\pm 6.84}$ |
| MSR-GC-PPO | $80.17_{\pm 3.69}$ | $68.82_{\pm 10.84}$ |

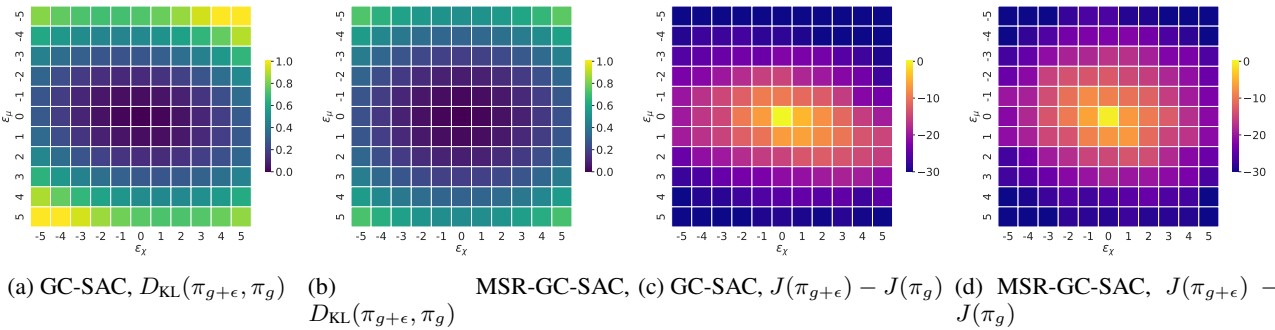

(a) GC-SAC, $D_{\mathrm{KL}}(\pi_{g+\epsilon}, \pi_g)$  (b) MSR-GC-SAC, $D_{\mathrm{KL}}(\pi_{g+\epsilon}, \pi_g)$  (c) GC-SAC, $J(\pi_{g+\epsilon}) - J(\pi_g)$  (d) MSR-GC-SAC, $J(\pi_{g+\epsilon}) - J(\pi_g)$

*Figure 18.* Policy discrepancy and return gap between policies for adjacent desired goals. Results come from experiments on **VVC** with **GC-SAC** and **MSR-GC-SAC**. The interpretation of the coordinate axes and the data collection methods are analogous to those described in Fig. 1.

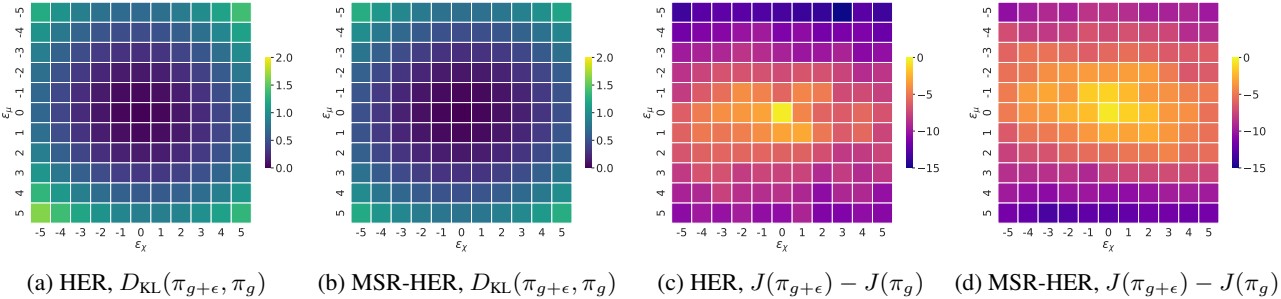

(a) HER, $D_{\mathrm{KL}}(\pi_{g+\epsilon}, \pi_g)$  (b) MSR-HER, $D_{\mathrm{KL}}(\pi_{g+\epsilon}, \pi_g)$  (c) HER, $J(\pi_{g+\epsilon}) - J(\pi_g)$  (d) MSR-HER, $J(\pi_{g+\epsilon}) - J(\pi_g)$

*Figure 19.* Policy discrepancy and return gap between policies for adjacent desired goals. Results come from experiments on **VVC** with **HER** and **MSR-HER**. The interpretation of the coordinate axes and the data collection methods are analogous to those described in Fig. 1.

the Reach task.

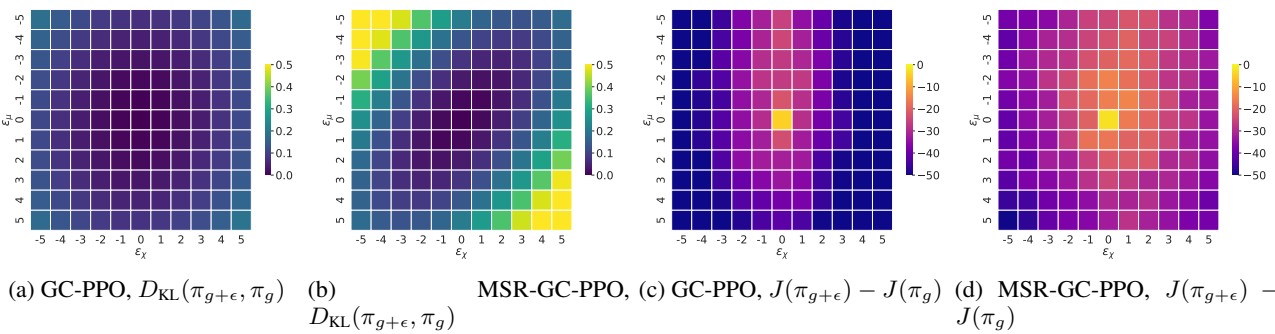

(a) GC-PPO, $D_{\mathrm{KL}}(\pi_{g+\epsilon}, \pi_g)$ (b) MSR-GC-PPO, (c) GC-PPO, $J(\pi_{g+\epsilon}) - J(\pi_g)$ (d) MSR-GC-PPO, $J(\pi_{g+\epsilon}) - J(\pi_g)$
$D_{\mathrm{KL}}(\pi_{g+\epsilon}, \pi_g)$

*Figure 20.* Policy discrepancy and return gap between policies for adjacent desired goals. Results come from experiments on **VVC** with **GC-PPO** and **MSR-GC-PPO**. The interpretation of the coordinate axes and the data collection methods are analogous to those described in Fig. 1.

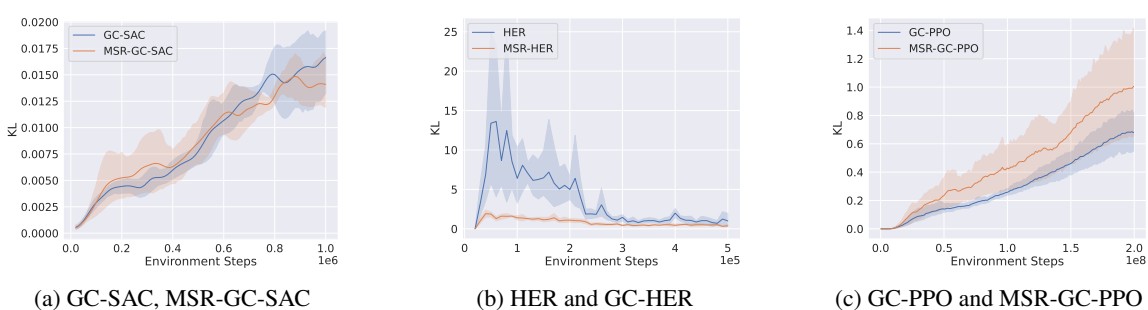

(a) GC-SAC, MSR-GC-SAC      (b) HER and GC-HER      (c) GC-PPO and MSR-GC-PPO

*Figure 21.* Trends of policy discrepancy between policies for adjacent desired goals during training on **VVC**. Results come from algorithms over 5 random seeds. Note that the policy discrepancy is measured by noise $\epsilon \sim [-0.1 \cdot \delta, 0.1 \cdot \delta]$.

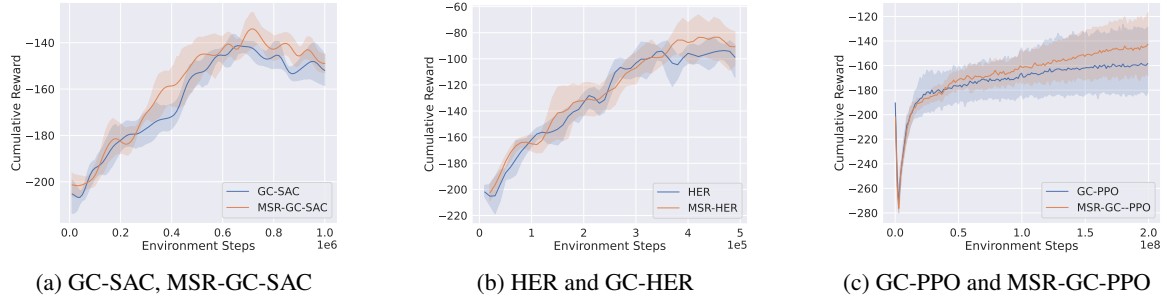

(a) GC-SAC, MSR-GC-SAC      (b) HER and GC-HER      (c) GC-PPO and MSR-GC-PPO

*Figure 22.* Cumulative rewards of policy during training on **VVC**. Results come from algorithms over 5 random seeds.

*Table 9.* Success rate of adjacent goals around an achievable goal and overall success rate of policy for different algorithms on **VVC**. The mean and variance are shown over 5 random seeds.

| Algorithms | Adjacent Success Rate (%) | Success Rate (%) |
|---|---|---|
| GC-SAC | $71.72_{\pm 8.87}$ | $20.86_{\pm 7.75}$ |
| MSR-GC-SAC | $74.32_{\pm 7.46}$ | $22.28_{\pm 6.84}$ |
| HER | $89.69_{\pm 3.03}$ | $62.84_{\pm 12.12}$ |
| MSR-HER | $90.97_{\pm 3.59}$ | $67.94_{\pm 13.03}$ |
| GC-PPO | $54.22_{\pm 8.02}$ | $7.62_{\pm 8.06}$ |
| MSR-GC-PPO | $69.11_{\pm 5.65}$ | $26.52_{\pm 33.64}$ |

*Table 10.* Comparison of TRPO, Offline-to-Online and MSR.

| Algorithm | Optimization Objective |
|---|---|
| TRPO | $E_{g \sim p_{dg}} KL[\pi_{\theta_{t-1}}(\cdot|\cdot, g) \| \pi_{\theta_t}(\cdot|\cdot, g)]$ |
| Offline-to-Online | $E_{g \sim p_{dg}} KL[\pi_0(\cdot|\cdot, g) \| \pi_{\theta_t}(\cdot|\cdot, g)]$ |
| MSR | $E_{g \sim p_{dg}, \epsilon \sim (-\epsilon', \epsilon')} KL[\pi_{\theta_t}(\cdot|\cdot, g) \| \pi_{\theta_t}(\cdot|\cdot, g + \epsilon)]$ |

## E. Comparison of MSR with Other KL-Based Policy Optimization Methods

Table 10 delineates the optimization objectives of TRPO (Schulman et al., 2015), Offline-to-Online (Baker et al., 2022; Gong et al., 2024b), and MSR under multi-goal settings. It can be observed that:

- TRPO aims to prevent the policy from changing too much during optimization, thus it achieves this objective by constraining the KL divergence between policies in consecutive training iterations.

- The Offline-to-Online method aims to prevent the policy from forgetting the knowledge learned offline during online learning, by constraining the KL divergence between the offline-learned policy, $\pi_0$, and the policy currently being learned online, $\pi_{\theta_t}$.

- MSR, on the other hand, seeks to enhance the policy's continuity of goal-achievement ability by limiting the divergence between policies corresponding to adjacent goals.

In summary, although all these methods utilize KL divergence, from the perspective of optimization objectives, MSR is fundamentally distinct from other KL-based methods.

