# OpenReview forum: "Improving the Continuity of Goal-Achievement Ability via Policy Self-Regularization for Goal-Conditioned Reinforcement Learning"
_ICML.cc/2025/Conference — ICML 2025 poster_

### Official Review · Reviewer_F6UM · 2025-03-09

**Overall Recommendation:** 3

**Summary:**

This paper addresses the issue of discontinuity in goal-achievement capabilities in Goal-Conditioned Reinforcement Learning (GCRL) algorithms. First, this paper theoretically proof that reusing successful trajectories can help achieve adjacent goals, but policy discrepancies must be controlled to avoid performance degradation. To tackle this, this paper proposes a Margin-Based Policy Self-Regularization (MSR) approach, which constrains policy differences between adjacent goals within a minimal threshold. The approach is evaluated on two robotic arm control tasks and an aircraft control task, demonstrating improved continuity in goal-achievement ability.

**Claims And Evidence:**

yes.

**Essential References Not Discussed:**

Yes. It could be good if this paper can discuss the relationship between MSR and TRPO, PPO methods in designing the MSR loss.

**Experimental Designs Or Analyses:**

Yes. This paper conducts performance analysis  on different GCRL algorithms.

**Methods And Evaluation Criteria:**

yes. Evaluation metrics include policy discrepancy (KL divergence), return gap, and cumulative rewards, which are appropriate for the problem.

**Other Comments Or Suggestions:**

1. It would be beneficial for the paper to analyze whether the policy regularization introduced by MSR adds significant computational overhead during inference.
2. The paper should consider incorporating a convergence analysis of the proposed MSR method. As shown in Figure 4(b), the cumulative reward appears not to have converged, raising concerns about the stability and reliability of the learning process. A detailed convergence analysis would provide insights into the long-term performance and robustness of the MSR approach.

**Other Strengths And Weaknesses:**

Strengths:
1. The paper effectively identifies and addresses the challenge of discontinuity in goal-achievement capabilities within GCRL. This clear motivation underpins the design of MSR.
2. The authors present a robust theoretical foundation for their work, enhancing the credibility and depth of the proposed MSR method.
3.  The inclusion of a thorough ablation study strengthens the paper by demonstrating the impact of various components of the MSR approach.

Weaknesses:
1. The MSR method introduces at least three key hyperparameters ($\lambda$, $\beta$, $\epsilon$) that need careful tuning. This requirement may increase the complexity of applying the approach across different environments and tasks.
2. The ablation study indicates that the optimal value for the hyperparameter λ is 0.001, which seems quite small. This raises questions about the significance of the MSR loss contribution. Additionally, the lack of explanation for Figure 7's results makes it challenging to understand the limited impact observed. A comparative analysis would be beneficial to elucidate this point.
3. Theoretical Equations (1) and (2) present symmetric upper and lower bounds. There is a concern that these bounds might be too loose in practical settings, potentially affecting the precision of the theoretical guarantees.

**Questions For Authors:**

1. The theoretical results assume a distance-based reward. Would MSR still work for other reward types?
2. Does policy self-regularization introduce significant training overhead? How does MSR impact convergence speed?
3. Why the optimal value of λ is quite small? Could the authors add some analysis?
4. Is Figure 4(b) converged?
5. Equations (1) and (2) have symmetric upper and lower bounds, are the bounds tight?

**Relation To Broader Scientific Literature:**

The paper positions itself within GCRL literature.

**Theoretical Claims:**

Yes. The theoretical results are well-structured and logically sound. Theorem 3.3 correctly bounds cumulative rewards in terms of KL divergence, providing rigorous justification for the method.

---

> ### Author Rebuttal · Authors · 2025-03-29
>
> `... discuss the relationship between MSR and TRPO, PPO methods in designing the MSR loss.`
>
> We denote the policy at the t-th iteration as $\pi_{\theta_t}$, with $\theta$ denoting policy parameters. $\pi_{\theta_t}(\cdot|s,g)$ maps state $s$ and goal $g$ to an action distribution.
>
> TRPO and PPO's key principle is to limit policy updates by constraining the KL divergence between policies in consecutive training iterations, $E_{g\sim p_{dg}}D_{KL}(\pi_{\theta_{t-1}} (\cdot|\cdot,g),\pi_{\theta_t}(\cdot|\cdot,g))$. In contrast, MSR restricts the policy's action distribution variation across goals within the same iteration, $E_{g\sim p_{dg},\epsilon \sim(-\epsilon',\epsilon')}D_{KL}(\pi_{\theta_t}(\cdot|\cdot,g),\pi_{\theta_t}(\cdot|\cdot,g+\epsilon))$.
>
> In essence, TRPO, PPO, and MSR use KL divergence for policy constraints but focus on distinct optimization objectives. This clarification will be added to our paper.
>
> `Would MSR still work for other reward types?`
>
> MSR is effective for all reward types:
>
> 1. Theorem 3.4 makes no assumptions about the reward function. The return gap remains bounded, albeit with a complex expression. For clarity, we derived a simpler bound under the distance-based reward, facilitating understanding of the relationship between $|J(\pi_{g_1})-J(\pi_{g_2})|$ and $d_{KL}(\pi_{g_1}(\cdot|s),\pi_{g_2}(\cdot|s))$.
> 2. In our experiments, we used both distance-based rewards (in Reach and VVC) and sparse rewards (in Push). Results demonstrate MSR's effectiveness across these different reward settings.
>
> `Does MSR introduce significant training overhead? How does MSR impact convergence speed?`
>
> MSR adds training overhead without influencing inference, due to added regularization only in policy optimization. For on-policy RL, where the main cost is environment sampling, MSR's effect is negligible, with only a 2.14% increase. In off-policy RL, which leverages replay buffer data, the impact is more pronounced: a 21.14% increase for MSR-GC-SAC and 16.02% for MSR-HER. Figs. 13, 17, and 21 reveal MSR's convergence rate is on par with baselines. Overall, despite the slight added training overhead, MSR markedly improves policy continuity in achieving goals.
>
> ||GC-PPO|MSR-GC-PPO|GC-SAC|MSR-GC-SAC|HER|MSR-HER|
> |:-:|:-:|:-:|:-:|:-:|:-:|:-:|
> |Wall Clock Time (s)|3498|3573|2621|3175|5096|5912|
>
> _**Note**: Results come from PPO (1e7 steps), SAC (1e5 steps), and HER (1e5 steps) on VVC._
>
> `Why the optimal value of λ is quite small?`
>
> We address your concerns from three aspects:
>
> 1. MSR is designed to lower $D_{KL} (\pi_{g_1}(\cdot|s), \pi_{g_2}(\cdot|s))$ without being excessively small. While 0.001 may seem trivial, our ablation studies confirm it sufficiently meets our objectives.
> 2. We've included a [plot](https://github.com/anonymouslinks/anonymous/blob/main/her_losses_lambda_1e-3_in_training.png) showing training loss trends with $\lambda = 0.001$. MSR loss is comparable to entropy and critic losses and an order of magnitude less than actor loss, demonstrating its impact on policy optimization despite the small $\lambda$.
> 3. Small coefficients for auxiliary losses are standard in RL. The primary goal is to maintain the RL objective's dominance, with other losses in a supporting role. For instance, entropy loss in PPO typically falls within $[10^{-5}, 10^{-2}]$, as seen in the [Stable Baselines3 Zoo](https://github.com/DLR-RM/rl-baselines3-zoo/blob/master/hyperparams/ppo.yml).
>
> `Are the bounds of Eq.(1) and (2) tight?`
>
> Eq.(1) and (2) can be unified as $|J(\pi_{g+\epsilon})-\frac{1}{1-\gamma}\sum_{(s,a)}\rho_{\pi_g}(s,a)r(s,a,g+\epsilon)| \le |\frac{2\sqrt{2}R_{\text{max}}}{(1-\gamma)^2}\sqrt{D_{KL}(\pi_{g+\epsilon}(\cdot|s),\pi_g(\cdot|s))}|$. We examine the tightness of this bound up to a constant with [a particular MDP instance](https://github.com/anonymouslinks/anonymous/blob/main/MDP.png). In the case of $g_1$ with policy $\pi_{g_1}(\cdot|s_0)=(0.6,0.4)$:
>
> 1. For $\pi_{g_2}$ identical to $\pi_{g_1}$, the left-hand side equals $\frac{0.2\gamma}{1-\gamma}$, while the right-hand side is $\frac{0.4\gamma}{1-\gamma}$, showing a constant factor difference.
> 2. When $\pi_{g_2}$ is slightly different, with $\pi_{g_2}(\cdot|s_0)=(0.55,0.45)$, the left-hand side becomes $\frac{0.15\gamma}{1-\gamma}$, and the right-hand side is $\frac{0.4\gamma + 0.2}{1-\gamma}$, again differing by a constant factor.
>
> We will include the detailed proof in the Appendix of manuscript.
>
> `(1) Is Figure 4(b) converged?. (2) The lack of explanation for Figure 7's results ...`
>
> Thank you for your suggestion. We have increased the training steps for HER and MSR-HER on Push and updated Fig.4 and added a convergence analysis in the manuscript. Fig.7 illustrates that a larger MSR strength $\lambda$ results in a stronger constraint on $D_{KL}(\pi_{g+\epsilon}(\cdot|s),\pi_g(\cdot|s))$ (values approach 0). We will provide a detailed analysis of the experimental data and corresponding conclusions in the manuscript.

---

### Official Review · Reviewer_Lfqn · 2025-03-12

**Overall Recommendation:** 3

**Summary:**

Reaching adjacent goals utilizing the same policy is non-trivial due to the limited robustness of policy improvement. The paper studies discontinuity in goal-achievement observed in Goal-Conditioned Reinforcement Learning (GCRL). Theoretically, the paper identifies constraints between goal reaching policies of the current goal and adjacent goal in order strike a balance between similarity and policy diversity which can hinder policy performance. This leads to the creation of Margin-Based Policy Self-Regularization (MSR) which regularizes a given RL policy with the KL constraint between the current and adjacent goal reaching policies. MSR can be combined with any RL policy and on the evaluated benchmarks, the method shows performant goal reaching ability while minimizing the KL constraint.

**Claims And Evidence:**

Please refer to strengths and weaknesses.

**Essential References Not Discussed:**

Please refer to strengths and weaknesses.

**Experimental Designs Or Analyses:**

Please refer to strengths and weaknesses.

**Methods And Evaluation Criteria:**

Please refer to strengths and weaknesses.

**Other Comments Or Suggestions:**

NA

**Other Strengths And Weaknesses:**

### Strengths

* The paper is well written and easy to follow.
* Authors have presented the theoretical results in an intuitive manner.

### Weaknesses

* **Theory To Practice**: My main concern is the connection of theoretical results to the practical algorithm. Theoretical claims present that previously trained policies can reach similar and adjacent goals while obeying similarity constraints, ie- the goal reaching policies are similar but not the same. However, I am struggling to understand how this connects to the empirical method of regularizing the policy with the divergence between policies. Intuitively, we want to balance between the similarity between the two policies but the method regularizes the RL objective using this as a constraint. It would be more suitable to automatically adjust the policy using a distillation scheme or perhaps learn a temperature parameter. Could the authors explain on how clipping the KL term connects to theoretical results?
* **Empirical Evaluation**: The current empirical evaluation is limited to only a few tasks and evaluation of $J(\pi)$ and $\text{KL}(\pi_{g}||\pi_{g+\epsilon})$ as optimization metrics. However, the paper does not answer the central question of how does MSR benefit the goal reaching abilities of RL algorithms? Does MSR help the policy reach farther goals? Does MSR benefit long-horizon reasoning of RL agents? How does MSR provide robustness to environmental perturbations? Another important aspect to validate is the empirical value of theoretical claims. Do the bounds obtained by authors hold for KL constraints? When $\text{KL}(\pi_{g}||\pi_{g+\epsilon}) = 0$, does the policy actually reach more adjacent goals? In its current form, the work only explores performance of goal reaching policy using prefixed metrics.
* **Contribution and Utility**: I am struggling to understand the novel contribution of the work and its utility for the RL community. How does the algorithm benefit RL algorithms since KL regularization is already a well known technique for policy improvement as well as GCRL. Besides, KL minimization is akin to likelihood maximization which raises the question of the new findings the paper contributes. Perhaps the main contribution lies in reaching adjacent goals via KL constraint as a distribution matching technique. In that case, it would be helpful for authors to study unseen environments or few-shot goal reaching improvement of the policy.

**Questions For Authors:**

Please refer to strengths and weaknesses.

**Relation To Broader Scientific Literature:**

Please refer to strengths and weaknesses.

**Theoretical Claims:**

Please refer to strengths and weaknesses.

---

> ### Author Rebuttal · Authors · 2025-03-31
>
> `W1: Theory To Practice`
>
> In our work, we aim to address the issue of ensuring that if a policy can achieve a goal $g$, it should also be capable of achieving goals in the vicinity of $g$, which we denote as $g+\epsilon$. From the perspective of cumulative rewards, our objective is to minimize $E_{g\sim p_{dg},\epsilon \sim(-\epsilon',\epsilon')}|J(\pi(\cdot|\cdot,g))-J(\pi(\cdot|\cdot,g+\epsilon))|$.
>
> Our method is informed by the following insights:
>
> 1. We show analytically that $E_{g\sim p_{dg},\epsilon \sim(-\epsilon',\epsilon')} |J(\pi(\cdot|\cdot,g))-J(\pi(\cdot|\cdot,g+\epsilon))|$ can be bounded by $KL[\pi(\cdot|\cdot,g)||\pi(\cdot|\cdot,g+\epsilon)]$, as concluded in Theorem 3.4, Corollary 3.5 and 3.6. This led us to consider constraining $KL[\pi(\cdot|\cdot,g)||\pi(\cdot|\cdot,g+\epsilon)]$ during policy optimization.
> 2. Theorem 3.3 suggests that the difference between $\pi(\cdot|\cdot,g)$ and $\pi(\cdot|\cdot,g+\epsilon)$ should not be entirely eliminated, implying that $KL[\pi(\cdot|\cdot,g)||\pi(\cdot|\cdot,g+\epsilon)]$ should not be optimized to be too small.
>
> Based on these findings, we develop the margin-based policy self-regularization method, as outlined in Eq.(6). The $KL$ term in the equation corresponds to the first point, and the max operation corresponds to the second point.
>
> `W2: Empirical Evaluation`
>
> We address your concerns from the following three aspects:
>
> **Firstly**, we believe that our experimental design aligns well with our theoretical derivations.
>
> In conjunction with our response to W1, our method aims to reduce $E_{g\sim p_{dg},\epsilon \sim(-\epsilon',\epsilon')}|J(\pi(\cdot|\cdot,g))-J(\pi(\cdot|\cdot,g+\epsilon))|$ by constraining $KL[\pi(\cdot|\cdot,g)||\pi(\cdot|\cdot,g+\epsilon)]$, thereby achieving the objective of enabling the policy to achieve not only $g$ but also goals in the vicinity of $g$. When designing our experiments, we sought to answer the following questions:
>
> 1. Can MSR reduce $KL[\pi(\cdot|\cdot,g)||\pi(\cdot|\cdot,g+\epsilon)]$?
> 2. whether a reduction in $KL[\pi(\cdot|\cdot,g)||\pi(\cdot|\cdot,g+\epsilon)]$ leads to a decrease in $E_{g\sim p_{dg},\epsilon \sim(-\epsilon',\epsilon')}|J(\pi(\cdot|\cdot,g))-J(\pi(\cdot|\cdot,g+\epsilon))|$?
> 3. can MSR enhance the policy's cumulative rewards, $J(\pi)$?
>
> The first two questions directly correspond to our theoretical derivations, while the third question assesses the practicality of our method, indicating that improving the continuity of the policy's goal-achievement can also enhance its ability to obtain cumulative rewards. Therefore, we consider our experimental design to be well-aligned with our theoretical development.
>
> **Secondly**, our focus is on improving policy continuity for goal achievement by reducing $E_{g\sim p_{dg},\epsilon \sim(-\epsilon',\epsilon')}|J(\pi(\cdot|\cdot,g))-J(\pi(\cdot|\cdot,g+\epsilon))|$, not on reaching farther goals, long-horizon reasoning, or robustness to environmental perturbations.
>
> **Thirdly**, our Section 4.4 ablation study with $\lambda=1$ shows $KL(\pi_g||\pi_{g+\epsilon})$ nearing 0 (Fig.7(a)), but $J(\pi)$ is lowest (Fig.7(b)). This observation aligns with Theorem 3.3, emphasizing the need to balance the reduction of $KL(\pi_g||\pi_{g+\epsilon})$ to maintain goal-achievement continuity without overly compromising performance.
>
> `W3: Contribution and Utility`
>
> The KL divergence is commonly used as a measure of distance to constrain policies in policy optimization. In the context of multi-goal settings, we compare MSR method with two KL-based policy constraint approaches:
>
> ||Optimization Objective|
> |:-:|:-:|
> |TRPO[1]|$E_{g\sim p_{dg}}KL[\pi_{\theta_{t-1}}(\cdot\vert\cdot,g)\Vert\pi_{\theta_t}(\cdot\vert\cdot,g)]$|
> |Offline-to-Online[2]|$E_{g\sim p_{dg}}KL[\pi_0(\cdot\vert\cdot,g)\Vert\pi_{\theta_t}(\cdot\vert\cdot,g)]$|
> |MSR|$E_{g\sim p_{dg},\epsilon \sim(-\epsilon',\epsilon')}KL[\pi_{\theta_t}(\cdot\vert\cdot,g)\Vert\pi_{\theta_t}(\cdot\vert\cdot,g+\epsilon)]$|
>
> 1. TRPO aims to prevent the policy from changing too much during optimization, thus it achieves this objective by constraining the KL divergence between policies in consecutive training iterations.
> 2. The Offline-to-Online method aims to prevent the policy from forgetting the knowledge learned offline during online learning, by constraining the KL divergence between the offline-learned policy, $\pi_0$, and the policy currently being learned online, $\pi_{\theta_t}$.
> 3. MSR, on the other hand, seeks to enhance the policy's continuity of goal-achievement ability by limiting the divergence between policies corresponding to adjacent goals.
>
> **In summary, although all these methods utilize KL divergence, from the perspective of optimization objectives, MSR is fundamentally distinct from other KL-based methods.**
>
> [1] Schulman, John, et al. Trust region policy optimization. ICML,2015.
>
> [2] Baker, Bowen, et al. Video pretraining (vpt): Learning to act by watching unlabeled online videos. NeurIPS,2022.

---

> > ### Comment · Reviewer_Lfqn · 2025-04-04
> >
> > I thank the authors for their response to my comments. After going through the authors' rebuttal and response to other reviewers, my concerns have been addressed and I have decided to raise my score. I thank the authors for their efforts.

---

### Official Review · Reviewer_xiNi · 2025-03-13

**Overall Recommendation:** 4

**Summary:**

In their present paper, the authors address an evident issue appearing in goal conditioned RL: discontinuity between control policies even in cases of adjacent goals, i.e. where their respective goals are only marginally separated by some distance \eps. The insights generated by a comprehensive analysis of the mentioned effect, which takes into account the KL divergence as a central measure to define policy discrepancy, are consequently turned into a new policy regularization method, MSR (Margin-Based Policy Self-Regularization), which can be attached to common off- and on-policy RL algorithms, such as SAC, HER and PPO, in order to improve not only the continuity properties between adjacent goal policies but also the overall policy performance (as measured by the total return value over all achievable goals). Based on three individual benchmark environments, the authors provide sound experimental support for their analytical claims and demonstrate the practical applicability of their method.

**Claims And Evidence:**

All claims in the present work are supported by both theoretical arguments, mathematical proofs and experimental evidence. All terms and pre-requisite are sufficiently introduced and explained, as there also is a good balance of showing essential parts of their analysis in the main text and providing further details in the appendix. The authors present a thorough experimental analysis of their central hypotheses. My only remark is that a higher number of random seed varying trials would improve their statistical evaluation of the experimental results, as the current number (5 trials) yields relatively large statistical error bars on the individual KPIs of Table 1.

**Essential References Not Discussed:**

The authors mainly discuss the relation to GCRL as applied to model-free policy-gradient and/or Q-value („critic“ function) related research. What is somehow missing is the relation to model-based RL methods, which are also strongly used in connection with multi-objective and goal-conditioned RL tasks, such as

- Marc P. Deisenroth and Dieter Fox. Multiple-target reinforcement learning with a single policy. ICML 2011 Workshop on Planning and Acting with Uncertain Models, 2011
- M. Weber, P. Swazinna, D. Hein, S. Udluft and V. Sterzing, "Learning Control Policies for Variable Objectives from Offline Data," 2023 IEEE Symposium Series on Computational Intelligence (SSCI)

**Experimental Designs Or Analyses:**

To my opinion, the authors have chosen a reasonable selection of benchmark comparisons, including representatives of both off-policy and on-policy GCRL SOTA algorithms. The simplicity of extending each candidate with the proposed regularization term helps to have a clear structured comparison between methods with and without the MSR extension. The visual illustrations are generally well-chosen and supportive to understand the most relevant effect.

However, the authors must improve the readability of axis labels and (color) legends used in their figure, as they are currently far too small or sometimes missing/unreadable at all. This is a mandatory issue for any final version provided if accepted for publication. Some of the plots in the main text are borderline small, so I strongly recommend finding a solution to make central results more visible.

**Methods And Evaluation Criteria:**

I don’t find any major flaw or concern in the continuity analysis for adjacent goal conditioned policies, as provided in Section 3. The proposed method, MSR, is reasonably introduced as a direct consequence of the analytical results and, as the authors claim, can be used to augment essentially any GCRL policy algorithm as an addition regularization term added to the reward definition. It authors thereby introduce a reasonable and relatively low number of (hyper-)parameters to enable fine-tuning of their regularization term. A sufficiently detailed ablation study for all mentioned hyper-parameters is provided as part of the experimental evaluation.

The evaluation criteria chosen (essentially three metrics shown in columns of Table 1) are reasonably chosen to highlight and support the claims and conclusions of the authors. As mentioned in the previous section, it appears that all results reported in Table 1 still come with a comparably large statistical error. In a few cases, the error become large enough to weaken the substance of conclusion taken by the authors, as they may also be result of random fluctuations. Reducing this error by increasing the number of random seed trials would help to add significance to the results and should be considered in the final version of the paper.

**Other Comments Or Suggestions:**

I repeat from previous section:
The authors must improve the readability of axis labels and (color) legends used in their figure, as they are currently far too small or sometimes missing/unreadable at all. This is a mandatory issue for any final version provided if accepted for publication. Some of the plots in the main text are borderline small, so I strongly recommend finding a solution to make central results more visible.

**Other Strengths And Weaknesses:**

Strengths:
————
The authors find a clear language and easy-to-follow structure to motivate the underlying issue with discontinuity in relation to adjacent goal conditioned control policies. The paper appears well written and succeeds in providing a clear relation of claims and evidence.

Comments/Remarks:
————————

l. 255 (right column): If indeed GC-PPO fails entirely at solving the general RL tasks in the mentioned benchmarks, it is questionable to consider any of those results to draw conclusions about the (dis-)continuity properties and total return accumulation (as partly admitted by the authors). Maybe those results should be marked or put into brackets in Table 1, to prevent false conclusions.

**Questions For Authors:**

None

**Relation To Broader Scientific Literature:**

The authors were generally short on introducing related works in Section 5, however, do provide a sound embedding of their work into the context of GCRL. They underline that, according to the best of their knowledge, the paper presents the first to focus on the continuity issue in context of GCRL. I personally cannot judge whether this is true but I also have no counter-example to object their claim.

**Theoretical Claims:**

I checked soundness and plausibility of all equations as part of the main text, and some (but not all) of the more detailed derivations provided in the appendix.

---

> ### Author Rebuttal · Authors · 2025-03-30
>
> We sincerely appreciate your valuable feedback and careful review of our paper. We address your concerns in the following:
>
> `My only remark is that a higher number of random seed varying trials would improve their statistical evaluation of the experimental results, as the current number (5 trials) yields relatively large statistical error bars on the individual KPIs of Table 1.`
>
> Thank you for your suggestion. We have increased the number of random seeds from 5 to 10, re-conducted the training and testing, and updated Table 1. The table below presents the results on VVC, showing that the variance of the obtained results has decreased to some extent.
>
> |Algorithms|$D_\text{KL}(\pi_{g+\epsilon}, \pi_g)$|$J(\pi_{g+\epsilon}) - J(\pi_g)$|$J(\pi)$|
> |:-:|:-:|:-:|:-:|
> |GC-SAC|0.37±0.12|-22.00±5.06|-138.20±14.16|
> |MSR-GC-SAC|0.33±0.09|-20.94±7.62|-132.09±8.06|
> |HER|0.65±0.22|-7.91±3.22|-69.21±12.87|
> |MSR-HER|0.58±0.16|-7.52±3.15|-64.73±13.55|
> |GC-PPO|0.08±0.08|-44.50±12.09|-169.04±25.57|
> |MSR-GC-PPO|0.17±0.21|-30.65±20.19|-146.64±28.05|
>
> Additionally, we have expanded our discussion to include the statistical significance of the results. Taking $D_\text{KL}(\pi_{g+\epsilon}, \pi_g)$ on VVC as an example, the table below presents the T-test's $\alpha$ values for the evaluation results of policies trained with 5 different random seeds between HER and MSR-HER. All the $\alpha$ values are below 0.05, indicating that MSR-HER can train policies with a smaller $D_\text{KL}(\pi_{g+\epsilon}, \pi_g)$ compared to HER.
>
> ||MSR-HER(1)|MSR-HER(2)|MSR-HER(3)|MSR-HER(4)|MSR-HER(5)|
> |:-:|:-:|:-:|:-:|:-:|:-:|
> |HER(1)|$2.4*10^{-5}$|$1.2*10^{-4}$|$3.3*10^{-11}$|$5.2*10^{-33}$|$8.5*10^{-13}$|
> |HER(2)||$7.1*10^{-16}$|$2.6*10^{-14}$|$5.4*10^{-4}$|$2.6*10^{-8}$|
> |HER(3)|||$2.9*10^{-4}$|$4.0*10^{-2}$|$2.3*10^{-4}$|
> |HER(4)||||$1.5*10^{-5}$|$2.5*10^{-6}$|
> |HER(5)|||||$7.6*10^{-7}$|
>
> _**Note**: We trained multiple HER and MSR-HER policies with distinct random seeds (numbers in parentheses indicate the seed index), each evaluated over 12100 episodes. As the T-test necessitates i.i.d. data for both groups being compared, and the test outcomes from different seeds do not belong to the same distribution, we can only conduct pairwise comparisons between the policies trained by HER and MSR-HER with varying random seeds._
>
> We are continuing to increase the number of random seeds to obtain more reliable experimental results. Once again, we thank you for your valuable suggestion.
>
> `However, the authors must improve the readability of axis labels and (color) legends used in their figure, as they are currently far too small or sometimes missing/unreadable at all. This is a mandatory issue for any final version provided if accepted for publication. Some of the plots in the main text are borderline small, so I strongly recommend finding a solution to make central results more visible.`
>
> We sincerely thank the reviewer for highlighting the issue regarding the readability of our figure axis labels and legends. To enhance the readability of all figures, we have taken the following steps:
>
> 1. We have increased the font size of all axis labels and legends to ensure they are easily readable.
> 2. For figures utilizing color legends, we have not only enlarged the text but also ensured that the color distinctions are more pronounced and clearly labeled.
>
> ` If indeed GC-PPO fails entirely at solving the general RL tasks in the mentioned benchmarks, it is questionable to consider any of those results to draw conclusions about the (dis-)continuity properties and total return accumulation (as partly admitted by the authors). Maybe those results should be marked or put into brackets in Table 1, to prevent false conclusions.`
>
> As suggested, we have marked the results of GC-PPO in Table 1 with italics to indicate that these results should be interpreted with caution due to the baseline algorithm's performance limitations. Additionally, we have included a description in the title to explain this notation. Thank you once again for your valuable insights.

---

> > ### Comment · Reviewer_xiNi · 2025-04-04
> >
> > I appreciate the authors' response to my remaining concerns and questions. They have satisfyingly addressed my comments regarding limited statistics and quality of figures and captions. I keep up my recommendation to accept their work (4).

---

### Official Review · Reviewer_MjdV · 2025-03-14

**Overall Recommendation:** 4

**Summary:**

This paper presents a regularization technique to improve the capabilities of Goal-Conditioned Reinforcement Learning (GCRL) algorithms. The authors start by motivating the need for their approach and presented prelimaries about GCRL. Next, the authors present a cohesive theoretical analysis displaying that modifying a policy toward a goal variant may achieve higher returns, and that the extent of the modification should not be excessive. Following, the authors present the main regularization technique: Margin-Based Policy Self-Regularization. The experiments in Section 4 are conducted across three different settings, two in a robotic control domain and one in aircraft control. Key takeaways include that MSR regularization 1) can enhance cumulative rewards for the policy while reducing the discrepancy between policies, and 2) The importance weight for the MSR objective must be chosen carefully.



#### Post-Rebuttal
The author's rebuttal includes an example of an improved caption, which will benefit the paper, and a statistical analysis of the results, which will improve the validity of the experimental findings. Both of these improvements should be included in the final manuscript.

**Claims And Evidence:**

Yes, the claims are supported by sufficient evidence.

**Essential References Not Discussed:**

None.

**Experimental Designs Or Analyses:**

The experimental design within the main paper was read carefully.

**Methods And Evaluation Criteria:**

The proposed methods and evaluation make sense for the problem at hand.

**Other Comments Or Suggestions:**

None.

**Other Strengths And Weaknesses:**

Strengths:
+ The proposed approach is backed by theoretical analysis.
+ There is an abundance of results supporting the method. The authors do a good job of displaying the policy discrepancy and cumulative rewards side-by-side.


Weaknesses:
- Many of the figure captions could be improved.
- It would be beneficial to verify that the claims made in Section 4 (that are based on Table 1) are valid with respect to statistical significance.

**Questions For Authors:**

Please respond to my weaknesses noted above.

**Relation To Broader Scientific Literature:**

This paper presents an important theoretical analysis that can benefit GCRL algorithms and inspire future research in this area.

**Theoretical Claims:**

Section 3 was read carefully. Proofs in the appendix were not verified in-depth.

---

> ### Author Rebuttal · Authors · 2025-03-30
>
> Thank you for your thorough review and valuable feedback on our work. We address your concerns in the following:
>
> `W1: Many of the figure captions could be improved.`
>
> We greatly appreciate your suggestions. In accordance with the principles of accuracy and conciseness, we have re-formulated the titles of all figures and tables. Taking Figure 6 as an example, which illustrates the policy discrepancy between adjacent desired goals, $D_\text{KL}(\pi_{g+\epsilon}, \pi_g)$, under various $\beta$ settings, supporting our ablation analysis on $\beta$, we have revised its title from:
>
> *"Ablation study on $\beta$. $D_\text{KL}(\pi_{g+\epsilon}, \pi_g)$ between policies for adjacent desired goals. The interpretation of the coordinate axes and the data collection methods are analogous to those described in Fig.1. Results come from MSR-GC-SAC on Reach over 5 random seeds."*
>
> to:
>
> *"Policy discrepancy between adjacent desired goals, $D_\text{KL}(\pi_{g+\epsilon}, \pi_g)$, under different $\beta$ settings. The meanings of the coordinate axes and the evaluation methods are consistent with those in Fig.1. Results are derived from MSR-GC-SAC on Reach across 5 random seeds."*
>
> `W2: It would be beneficial to verify that the claims made in Section 4 (that are based on Table 1) are valid with respect to statistical significance.`
>
> We appreciate your suggestion. Taking $D_\text{KL}(\pi_{g+\epsilon}, \pi_g)$ on VVC as an example, we have conducted a statistical significance analysis between HER and MSR-HER. The table below presents the T-test's $\alpha$ values for the evaluation results of policies trained with 5 different random seeds. All the $\alpha$ values are below 0.05, indicating that MSR-HER can train policies with a smaller $D_\text{KL}(\pi_{g+\epsilon}, \pi_g)$ compared to HER.
>
> ||MSR-HER(1)|MSR-HER(2)|MSR-HER(3)|MSR-HER(4)|MSR-HER(5)|
> |:-:|:-:|:-:|:-:|:-:|:-:|
> |HER(1)|$2.4*10^{-5}$|$1.2*10^{-4}$|$3.3*10^{-11}$|$5.2*10^{-33}$|$8.5*10^{-13}$|
> |HER(2)||$7.1*10^{-16}$|$2.6*10^{-14}$|$5.4*10^{-4}$|$2.6*10^{-8}$|
> |HER(3)|||$2.9*10^{-4}$|$4.0*10^{-2}$|$2.3*10^{-4}$|
> |HER(4)||||$1.5*10^{-5}$|$2.5*10^{-6}$|
> |HER(5)|||||$7.6*10^{-7}$|
>
> _**Note**: We trained multiple HER and MSR-HER policies with distinct random seeds (numbers in parentheses indicate the seed index), each evaluated over 12100 episodes. As the T-test necessitates i.i.d. data for both groups being compared, and the test outcomes from different seeds do not belong to the same distribution, we can only conduct pairwise comparisons between the policies trained by HER and MSR-HER with varying random seeds._
>
> Additionally, we noted the large variance in the results of Table 1. We have increased the number of random seeds from 5 to 10 and re-conducted the training and testing, updating Table 1. The updated table for the VVC shows a reduction in variance for the results.
>
> |Algorithms|$D_\text{KL}(\pi_{g+\epsilon}, \pi_g)$|$J(\pi_{g+\epsilon}) - J(\pi_g)$|$J(\pi)$|
> |:-:|:-:|:-:|:-:|
> |GC-SAC|0.37±0.12|-22.00±5.06|-138.20±14.16|
> |MSR-GC-SAC|0.33±0.09|-20.94±7.62|-132.09±8.06|
> |HER|0.65±0.22|-7.91±3.22|-69.21±12.87|
> |MSR-HER|0.58±0.16|-7.52±3.15|-64.73±13.55|
> |GC-PPO|0.08±0.08|-44.50±12.09|-169.04±25.57|
> |MSR-GC-PPO|0.17±0.21|-30.65±20.19|-146.64±28.05|
>
> Once again, we appreciate your suggestion and will include the discussion on statistical significance in our manuscript.

---

### Decision · Program_Chairs · 2025-05-01

**Decision:**

Accept (poster)

**Comment:**

The paper introduces a theoretically motivated policy self-regularization method to improve continuity in goal-conditioned RL. Reviewers praised the clear theoretical analysis and empirical validation. Minor concerns about statistical significance and figure clarity were adequately addressed in rebuttal. The proposed approach is original and broadly applicable.